# Soil Bulk Density, Aggregates, Carbon Stabilization, Nutrients and Vegetation Traits as Affected by Manure Gradients Regimes Under Alpine Meadows of Qinghai–Tibetan Plateau Ecosystem

**DOI:** 10.3390/plants14101442

**Published:** 2025-05-12

**Authors:** Mahran Sadiq, Nasir Rahim, Majid Mahmood Tahir, Aqila Shaheen, Fu Ran, Guoxiang Chen, Xiaoming Bai

**Affiliations:** 1College of Grassland Science, Gansu Agricultural University, Lanzhou 730070, China; khanmahran420@gmail.com (M.S.); ranfu321@gmail.com (F.R.); guoxch66@gmail.com (G.C.); 2Department of Soil and Environmental Sciences, University of Poonch, Rawalakot 12350, Pakistanaqeelaupr@gmail.com (A.S.); 3Key Laboratory of Grassland Ecosystem, Ministry of Education, Lanzhou 730070, China

**Keywords:** alpine meadow, carbon fractions, carbon stabilization, degradation, forage quality, manures, soil properties

## Abstract

Climate change and overgrazing significantly constrain the sustainability of meadow land and vegetation in the livestock industry on the Tibetan–Plateau ecosystem. In context of climate change mitigation, grassland soil C sequestration and forage sustainability, it is important to understand how manure regimes influence SOC stability, grassland soil, forage structure and nutritional quality. However, the responses of SOC fractions, soil and forage structure and quality to the influence of manure gradient practices remain unclear, particularly at Tianzhu belt, and require further investigation. A field study was undertaken to evaluate the soil bulk density, aggregate fractions and dynamics in SOC concentration, permanganate oxidizable SOC fractions, SOC stabilization and soil nutrients at the soil aggregate level under manure gradient practices. Moreover, the forage biodiversity, aboveground biomass and nutritional quality of alpine meadow plant communities were also explored. Four treatments, i.e., control (CK), sole sheep manure (SM), cow dung alone (CD) and a mixture of sheep manure and cow dung (SMCD) under five input rates, i.e., 0.54, 1.08, 1.62, 2.16 and 2.70 kg m^−2^, were employed under randomized complete block design with four replications. Our analysis confirmed the maximum soil bulk density (BD) (0.80 ± 0.05 g cm^−3^) and micro-aggregate fraction (45.27 ± 0.77%) under CK, whilst the maximum macro-aggregate fraction (40.12 ± 0.54%) was documented under 2.70 kg m^−2^ of SMCD. The SOC, very-labile C fraction (C*frac*_1_), labile C fraction (C*frac*_2_) and non-labile/recalcitrant C fraction (C*frac*_4_) increased with manure input levels, being the highest in 2.16 kg m^−2^ and 2.70 kg m^−2^ applications of sole SM and the integration of 50% SM and 50% CD (SMCD), whereas the less-labile fraction (C*frac*_3_) was highest under CK across aggregate fractions. However, manures under varying gradients improved SOC pools and stabilization for both macro- and micro-aggregates. A negative response of the carbon management index (CMI) in macro-aggregates was observed, whilst CMI in the micro-aggregate fraction depicted a positive response to manure addition with input rates, being the maximum under sole SM addition averaged across gradients. Higher SOC pools and CMI under the SM, CD and SMCD might be owing to the higher level of soil organic matter inputs under higher doses of manures. Moreover, the highest accumulation of soil nutrients,, for instance, TN, AN, TP, AP, TK, AK, DTPA extractable Zn, Cu, Fe and Mn, was recorded in SM, CD and SMCD under varying gradients over CK at both aggregate fractions. More nutrient accumulation was found in macro-aggregates over micro-aggregates, which might be credited to the physical protection of macro-aggregates. Overall, manure addition under varying input rates improved the plant community structure and enhanced meadow yield, plant community diversity and nutritional quality more than CK. Therefore, alpine meadows should be managed sustainably via the adoption of sole SM practice under a 2.16 kg m^−2^ input rate for the ecological utilization of the meadow ecosystem. The results of this study deliver an innovative perspective in understanding the response of alpine meadows’ SOC pools, SOC stabilization and nutrients at the aggregate level, as well as vegetation structure, productivity and forage nutritional quality to manure input rate practices. Moreover, this research offers valuable information for ensuring climate change mitigation and the clean production of alpine meadows in the Qinghai–Tibetan Plateau area of China.

## 1. Introduction

Global climate change and overgrazing are leading to serious threats to animal husbandry, leading to land degradation and declines in meadow SOC storage, grassland soil quality as well as forage structure, nutritional quality and productivity [1]. However, the SOC is the largest C pool in the terrestrial biosphere, and its sequestration is important in environmental modulations globally [2]. Currently, there is rising SOC recognition because of its many-sided effect on soil characteristics, vegetation biodiversity, productivity and nutritional quality while also holding carbon dioxide (CO_2_) sink potential [3]. This increased concern underscores SOC’s significance in evaluating soil and forage quality [4]. The accumulation of SOC is a utility of the bond between C inputs in the form of manures as well as straws and the decomposition rate viz. soil C breakdown, as mediated by the environment (climatic conditions and soil type) [5,6]. The continuous addition of organic materials, for instance, crop straws, biogas residues and manures, is an effective way to increase SOC status [7,8]. Therefore, an improved understanding of the effect of manures on SOC stabilization, productivity and forage quality parameters is imperative for grassland management techniques, the mitigation of global warming practices and conservation efforts [9].

Environment change mainly caused by global warming has become a crucial problem [10]. Global warming raises soil CO_2_ fluxes due to the stimulation of the SOC oxidation rate [11]. These changes in soil–atmosphere CO_2_ fluxes have a direct impact on global environments [12]. SOC fractions have two major categories: active or labile C fractions and stable, non-labile or recalcitrant C fractions. Active SOC fractions can rapidly decompose and become part of nutrient cycling in a soil system, whilst the stable SOC fraction is valuable for alleviating soil CO_2_ emissions in ecosystems [13]. The oxidation rate of permanganate-oxidizable SOC fractions is rapid and more sensitive to environmental changes and land management techniques [14]. However, SOC is a soil quality measurement. An improved understanding of the knowledge of its active and non-active fractions under manure input rates might be used for land degradation identification and conservation and restoration practices [14,15].

The physical protection of the soil aggregate fractions determines SOC pool stabilization [16]. Minor pores of micro-aggregates are valuable for shielding SOC decomposition, whilst bigger pores in macro-aggregates help the flow of gas–water, which has optimistic significance for metabolic processes and bacterial activity [17,18]. Moreover, the greater pore structure results in inferior SOC physical protection and easy substrate access for microbes [19]. Therefore, soil macro-aggregate fractions naturally have a quicker turnover rate over micro-aggregate fractions, and SOC related with macro-aggregate fractions might be more sensitive to environment variations [20]. Soil manure addition is an operative way to enhance SOC sequestration and increase soil aggregate formation [21,22]. Usually, SOC changes are very slow and are difficult to assess because of their greatly related levels. Labile permanganate-oxidizable SOC fractions under manure practices have smaller turnover times in the conversion between stabilized organic matter and fresh plant residue [23]. Therefore, knowledge on the changes in permanganate-oxidizable SOC fractions in response to manure addition is essential. The influences of manure addition on soil’s physical characteristics are largely progressive [24,25]. Soil bulk density is a significant soil compaction parameter and depends on the mineral particle density, organic matter and the packing arrangement [26]. Earlier studies [27,28] illustrated that manure addition resulted in a decrease in BD over synthetic fertilizer. The addition of manure to soil also greatly impacts the structure [25], resulting in increased aggregate stability [29]. In a field trial, Zhanhui et al. [30] found that manure improved the weight percentage of aggregates and aggregate stability. Manure’s influence on the soil extent is probably dependent on manure rates as well as properties and environmental factors that influence the decomposition of manure.

Sustainable meadow productivity requires plant biomass removal through animal consumption or fodder harvest, which contains essential nutrients in its tissues [31]. The long-term continuous removal of forage for the purpose of animal husbandry causes a major depletion in soil nutrients, and replenishing soil nutrients with animal manure is one of the most common input strategies employed to return what was removed through harvest [32]. Manure can be used as a fertilizer source of macro- and micro-nutrients as well as decreasing the necessity for synthetic chemical fertilizers in production systems, which can be costly and have ecological drawbacks [33]. Animal manure contains both organic and inorganic forms of micro-nutrients (Mn, Fe, Zn, B and Cu) and macro-nutrients (N, P, K as well as other macro-nutrients) [32,33]. Animal manure, here outlined as sheep manure and cow dung, has been accepted as a promising practice (and a fundamental management system) that preserves and restores soil nutrients and reduces chemical dependency [34]. Manure fertilization can improve both soil macro- and micro-nutrients [35]. The use of manures has been shown to increase the yield and soil nutrients under a north-western Indian ecosystem [22,35,36].

Forage chiefly affects meat and milk production in animal husbandry. If vegetation does not achieve the demand in relation to both richness and dominance, it might influence livestock nutrition [37,38]. Therefore, for high-quality vegetation development and the search to increase forage yield, techniques, for example, manure addition, should be considered. The integration of organic manures is a well-planned system for vegetation production, and it delivers many benefits, for example, promoting biomass and forage quality and enhancing grassland use efficiency. Manure causes changes in the composition of plants [39]. This improved availability of nutrients might favor certain species of plants that are more reactive to these nutrients, chief to plant composition changes and a reduction in plant community species diversity [40]. Moreover, Schröder et al. [41] showed that manure application increases the grass aboveground biomass yield by 20–56% without fertilization control due to the nutrient availability for plant growth. Manure use as an organic fertilizer has an optimistic influence on biomass yield, though its influence on forage nutritional quality is uncertain. Stybnarová et al. [42] noted a noteworthy increase in forage nutritional quality parameters, whereas Simi’c et al. [43] observed no significant impact on vegetation nutritional quality aspects. In addition to the numerous confident influences of manure application, there are also adverse features to consider, for instance, strong odors [44] and irregular plant growth [45]. Moreover, inappropriate manure application can lead to overfertilization and the compaction of soil, both of which can harshly degrade water and soil quality [46]. It can also result in the contamination of forage, which worsens forage quality and can lead to condensed animal forage intake and poor livestock performance [47].

The alpine ecosystems of the Qinghai–Tibet Plateau are harsh yet delicate and fragile, pointedly fluctuating due to climate change [48]. The sensitive climatic conditions of the Qinghai–Tibet Plateau underscore the critical significance of understanding changes in SOC, nutrients, meadow biodiversity, productivity and quality for ecosystem and grassland sustainability. SOC plays a crucial role in regulating nutrient cycling, maintaining forage structure, production and quality in this delicate alpine area, characterized by alpine shrub, alpine meadow, shrubland–grassland ecotone and herbs [49]. Alpine ecosystems are chiefly vulnerable to environmental hazards, for instance, climate change and erosion, which can have superficial influences on C stabilization and environment resilience [50]. Consequently, broad knowledge in our understanding of carbon, nutrient changes at the aggregate level and forage traits under various manure input rates is vital for informed meadow management decisions, sustainable development and restoration techniques and grassland conservation efforts for animal husbandry in this vulnerable and distinctive landscape. Previous studies only focused on soil physicochemical properties or the vegetation community under grazing regimes [51,52], but there are only few studies available on SOC pools, stability and nutrients, especially at the soil aggregate level, vegetation biodiversity, productivity and forage nutritional quality under manure practices across varying input rates. Therefore, the aim of this study was to estimate how BD, aggregate size distributions, soil aggregate-associated SOC pools and their stabilization, nutrients, vegetation structure, biomass and forage nutritional quality were influenced by manures, such as sole sheep manure (SM), cow dung alone (CD) and a mixture of both 50% SM and 50% CD across varying gradients in an alpine meadow ecosystem. We hypothesized the following: (i) the soil aggregation and soil aggregate-associated SOC, nutrients, meadow vegetation structure, biomass and nutritional quality will display an increase with manure gradient addition; (ii) within manure gradient practices, we anticipate that the sheep manure addition under higher input rates will reveal higher levels of soil aggregation, aggregate-associated SO fractions, nutrients, forage structure, biomass and meadow community nutritional quality and exhibit a higher C management index, signifying more operative C storage and sustainable practice than other manure practices. Our exploration of manure addition could deliver valuable visions into how different manure gradients affect the SOC levels in meadows of Tianzhu, Gansu, China. Meadow grassland bureaucrats, legislators and herdsmen can use this information to make informed decisions about alpine meadow restoration planning and management practices that promote SOC storage, stabilization, grassland sustainability and efficiency of alpine meadow resources. The Chinese Government and monitoring bodies can use scientific evidence on C changes to plan effective rules that promote ecological grassland management and develop targeted policies to address climate change problems in the meadow ecosystem.

## 2. Results

### 2.1. Soil Bulk Density and Aggregate Size Distribution

Organic manures under different input rates significantly decreased soil BD values, as shown in Figure 1a. The BD values ranged between 0.69 ± 0.07 and 0.80 ± 0.05 g cm^−3^ in all applied treatments. BD was notably lower in plots amended with SMCD under varying gradients compared to plots treated with sole SM and CD alone with different gradients. Maximum BD was recorded under unfertilized control. Moreover, enhancing the manure input rates significantly decreased BD, and the lowest value was noted under 2.16 kg m^−2^ across all manure treatments. The SM and CD were statistically on par in terms of the BD average over all gradients. Our results demonstrated that BD followed the trend of CK > CD > SM > SMCD.

The soil aggregate proportion in all treatments decreased with increasing aggregate size as the micro-aggregate proportion was higher than the macro-aggregate fraction. Manures with different input rates pointedly influenced the aggregate size distribution. All manure practices under varying gradients significantly increased the soil macro-aggregate fraction. The integration of SM and CD had the highest value of macro-aggregates, which was followed by sole SM and then CK across gradients. SMCD application with 2.70 kg m^−2^ had the highest macro-aggregate fraction value compared with sole SM and CD application with different gradients and CK. Increasing manure levels notably increased the macro-aggregate fraction. The lowest soil macro-aggregate fraction was noted under the unamended control. Compared with CK, the organic manures SM, CD and SMCD increased the macro-aggregate fraction by 12.51, 11.64, 15.23% on average across varying manure input rates (Figure 1b). Sole sheep manure, cow dung alone and integrated SM and CD application with input rates depicted significant reductions in the micro-aggregate fraction. CK had a significantly maximal value of micro-aggregates over manure addition under different input rates. Organic manure SM, CD and SMCD decreased the micro-aggregate fraction by 4.67, 3.87, 4.28% on average across manure input rates. Our results showed that the micro-aggregate fraction followed the trend of SM < SMCD < CD < CK (Figure 1c).

### 2.2. SOC Concentration, SOC Fractions and Stabilization

Significant variation was noted in the SOC concentration and permanganate-oxidizable SOC fractions, for instance, C*frac*_1_, C*frac*_2_, C*frac*_3_ and C*frac*_4_, in both soil macro- and micro-aggregate fractions under different manure gradient practices (Table 1). The proportion of each organic C fraction relative to total SOC under short-term manure gradient regimes at both soil aggregate fractions is shown in Figure 2a–h. The concentrations of SOC content and SOC fractions were higher in the macro-aggregate fraction over the micro-aggregate fraction, irrespective of different manure gradient treatments.

For the soil macro-aggregate fraction, manure treatments with an input rate of 2.16 kg m^−2^ had the highest SOC followed by 1.62 kg m^−2^ gradient over other manure input rates and CK. The maximum SOC was observed under sole SM across varying gradients. Organic manures SM, CD and SMCD increased the SOC content by 17.04, 10.67 and 15.90%, respectively, compared to CK across varying manure input rates. Almost parallel with the soil aggregate fraction above, for the soil micro-aggregate fraction, manures with 2.16 kg m^−2^ input rate had maximum SOC, which was followed by 1.62 kg m^−2^ gradient compared with other manure gradients and CK. The highest SOC, 100 ± 2.4 g kg^−1^, was associated with the integration of the SM and CD treatment average over gradients. The SM, CD and SMCD showed significantly higher SOC content by 21.34, 16.58 and 22%, respectively, compared with CK across different applied gradients (*p <* 0.05). In the micro-aggregate fraction, SOC depicted the trend of SMCD > SM > CD > CK across the tested manure input rates. The lowest SOC concentration in both soil aggregate fractions was observed in CK treatment (Table 1).

For soil macro-aggregates, C*frac*_1_ ranged from 0.2 ± 8.83 to 0.5 ± 10.10 g kg^−1^, constituting about 8.58–10.02% of total SOC (Table 1 and Figure 2a). The sole manure application (SM and CD) and integration of SM and CD resulted in a significant increase in C*frac*_1_ over CK; meanwhile, the C*frac*_1_ pointedly increased with enhancing manure rates from 0.54 kg m^−2^ to 2.16 kg m^−2^ then decreased. The highest C*frac*_1_ concentration was related to the 2.16 kg m^−2^ input rate of SM, CD and SMCD treatments. SM application alone had the maximum C*frac*_1_ value, followed by SMCD across different gradients. The values for C*frac*_1_ to SOC under manure input rates demonstrated a contrary trend to the C*frac*_1_ concentration, as a higher proportion was observed under CK, and the lowest proportion was associated with the 2.16 kg m^−2^ input rate of tested manures. Equivalent with soil macro-aggregates, regarding soil micro-aggregates, the 2.16 kg m^−2^ manure input rate had higher C*frac*_1_ than other manure input rates. The interaction of 50% sheep manure and 50% cow dung (SMCD) showed the maximum 11.53 ± 0.5 g kg^−1^ C*frac*_1_ concentration, followed by sole SM application across different gradients. The SMCD, SM and CD significantly increased C*frac*_1_ by 28.11, 26.22 and 25.22%, respectively, compared to CK across manure gradients. The lowest C*frac*_1_ value in both soil aggregates was noted in CK. In micro-aggregates, C*frac*_1_ illustrated the trend of SMCD > SM > CD > CK across manure input rates, constituting about 10.98–13.24% of total SOC. The C*frac*_1_ to SOC proportions in micro-aggregates under manure input rates were similar to the C*frac*_1_ concentration, increasing with manure input rates and lowest under CK (Figure 2b).

On average, of the different organic manure input rates, the mixture of 50% SM and 50% CD management (SMCD) practice showed the highest 16.55 ± 0.7 g kg^−1^ C*frac*_2_ which was statistically on par with the SM alone treatment compared to CK in the soil macro-aggregate fraction. It was increased under SM, CD and SMCD by 6.98, 4.33 and 7.05%, respectively, over CK across gradients. It was almost similar to C*frac*_1_ as the highest under the 2.16 kg m^−2^ input rate compared with other manure gradients and CK, irrespective of diverse manures. C*frac*_2_ accounted for about 15–17.56% of total SOC, and its contribution was higher than other labile C*frac*_1_ and C*frac*_3_ fractions. Its proportion relative to total SOC was significantly higher in CK than manure treatments (Figure 2c). Concerning the micro-aggregate fraction, SM alone revealed the maximum 14.32 ± 0.6 g kg^−1^ C*frac*_2_ value, followed by SMCD amendment across varying gradients. Manure treatments followed the trend of SM > SMCD > CD > CK across gradients in the case of C*frac*_2_. C*frac*_2_ in micro-aggregates followed a pattern similar to C*frac*_2_ in macro-aggregates among all input rates, as the highest C*frac*_2_ was observed for the 2.16 kg m^−2^ manure input rate. The C*frac*_2_ contributed the major labile proportion, 12.64–16.27%, of total SOC at the micro-aggregate level, being highest under sole CD and lowest under SMCD (Figure 2d). The CK treatment had the minimum concentration of C*frac*_2_ in both investigated soil macro- and micro-aggregate fractions (Table 1).

In the case of soil macro-aggregates, the addition of sole and combined organic manures with different input rates resulted in significantly decreased C*frac*_3_ values over untreated CK. The integrated application of SM along with CD had the lowest 10.61 ± 0.7 g kg^−1^ C*frac*_3_ value, followed by sole SM compared to the 11.45 ± 0.4 g kg^−1^ value of CK across manure input rates. The SMCD treatment significantly decreased C*frac*_3_ by 7.91% compared to CK, 0.5% more than SM alone and 3.11% than sole CD. Increasing manure input levels from 0.54 kg m^−2^ to 2.16 kg m^−2^ resulted in a lower C*frac*_3_ concentration. C*frac*_3_ contributed about 9–13% of the proportion of total SOC, being highest under CK and lowest under SMCD (Figure 2e). Data on soil C*frac*_3_ in the micro-aggregate fraction demonstrated that the lowest 10.35 ± 0.7 g kg^−1^ C*frac*_3_ was noted under SMCD compared to other manures across different gradients. Manures followed the trend of SMCD < SM < CD < CK across gradients regarding C*frac*_3_. The C*frac*_3_ concentration was inversely proportional to increasing manure input doses from 0.54 kg m^−2^ to 2.16 kg m^−2^. Its contribution was about 8.56–13.42% of total SOC, being lowest under SMCD and highest under CK (Figure 2f). Interestingly, in soil macro-aggregates, the C*frac*_3_ concentration was highest under CK, whilst, in the case of soil micro-aggregates, its highest concentration was reported under a low input rate in CD practice (Table 1).

The C*frac*_4_ concentration in soil macro-aggregates followed a pattern similar to total SOC and C*frac*_1_ concentrations among all treatments, as the highest C*frac*_4_ was recorded under sole SM followed by SMCD across gradients. Organic manures SMCD, SM and CD increased C*frac*_4_ by 25.42, 27.60 and 17%, respectively, over CK across the tested gradients. Increasing manure input levels from 0.54 kg m^−2^ to 2.16 kg m^−2^ resulted in a rise in the C*frac*_4_ concentration. The 2.16 kg m^−2^ gradient of SM, CD and SMCD depicted better overall performance compared with other manure input rates. C*frac*_4_ constituted 59.59 to 67.20% of total SOC, highest under SMCD, followed by SM compared with CK (Figure 2g). SMCD had the highest C*frac*_4_ concentration in soil micro-aggregates, which was statistically on par with sole SM over CK across manure input rates. The C*frac*_4_ concentration followed the trend of SMCD > SM > CD > CK on average over manure input rates. C*frac*_4_ accumulation was higher in the 2.16 kg m^−2^ manure gradient over other manure input rates. The fraction of SOC as C*frac*_4_ in the micro-aggregates ranged from 60 to 66% (Figure 2h) and exhibited a pattern similar to C*frac*_4_ in macro-aggregates. The minimum value of C*frac*_4_ was observed in the unamended control for both aggregate fractions (Table 1).

In general, our results exhibit that C*frac*_2_ was the dominant labile C fraction in comparison with other labile C fractions (C*frac*_1_ and C*frac*_3_) under diverse meadow management practices. Moreover, in our study, C*frac*_4_ was the dominant SOC fraction compared to C*frac*_1_, C*frac*_2_ and C*frac*_3_ under manure application. These permanganate-oxidizable organic carbon fractions followed the trend of C*frac*_1_ < C*frac*_3_ < C*frac*_2_ < C*frac*_4_ in both soil macro- and micro-aggregate fractions, irrespective of different manure input rates.

The influence of sole SM, CD alone and the combined use of both 50% SM and 50% CD on carbon indices, specifically carbon pool index (CPI), lability (L), lability index (LI), carbon management index (CMI) and stability index (SI), is presented in Table 2. Mean carbon index values underscored the statistically significant differences among the treatments in both investigated soil aggregate fractions. Specifically, the sole SM treatment manifested the highest mean CPI (0.04 ± 1.17) in macro-aggregates and in micro-aggregates (0.04 ± 1.22), followed by SMCD, irrespective of manure input rates. CK registered the lowest CPI (0.00 ± 1.00) for both soil aggregate fractions. Initially, an ascending trend and, finally, descending trend in CPI values were seen with increasing manure input rates (Table 2), evident from the progression from 0.54 kg m^−2^ to 2.16 kg m^−2^ and then regression at 2.70 kg m^−2^ for both aggregate fractions. Data demonstrated that the SM alone technique had the significantly lowest (0.55 ± 0.02) L value in macro-aggregates, and SMCD had the lowest (0.02 ± 0.55) L value in micro-aggregates compared with CK. Manure input rates increased from 0.54 kg m^−2^ to 2.16 kg m^−2^, resulting in a reduction in the L value for both aggregate fractions (Table 2).

Different manures under input rates significantly (*p < 0.05*) reduced LI for both macro-aggregates and micro-aggregates compared with CK, except 2.70 kg m^−2^ of CD for micro-aggregates. For both aggregate fractions, CD treatment had a higher LI than SM and SMCD across varying gradients. In general, increasing manure input rates decreased the LI values for both explored aggregate fractions (Table 2).

In the macro-aggregate fraction, the CMI value of CK was greater than all manures under varying gradient-treated amendments. On average, of the gradients, the CMI of CD was highest in comparison with SM and SMCD. A descending trend was observed in CMI values with increasing manure input rates, except 2.70 kg m^−2^ of SMCD treatment. In the micro-aggregate fraction, a contrasting trend with soil macro-aggregates was recorded as different manure input rates significantly (*p* < 0.05) increased the CMI compared with CK. Across varying gradients, SM had the highest CMI, which was statistically on par with CD over CK. The CMI followed the trend of SM > CD > SMCD > CK. Overall, 2.70 kg m^−2^ of SM, CD and SMCD depicted better performance over the other investigated manure gradients (Table 2).

In the macro-aggregate fraction, sole SM had the maximum SI, which was followed by SMCD over CK across varying input rates. Compared with CK, the SM, CD and SMCD increased SI by 23,97, 15.75 and 22.60%, respectively, across varying manure input rates. The 2.16 kg m^−2^ of SM, CD and SMCD demonstrated a higher SI over the other tested input rates. In the micro-aggregate fraction, the integrated use of SM and CD had the highest SI, which was followed by sole SM treatment across manure gradients. In general, 2.70 kg m^−2^ of different applied manures, especially SMCD, illustrated better performance than the other investigated gradients. SI followed the trend of SMCD > SM > CD > CK across varying manure input rates. The control treatment in both aggregate fractions had the lowest SI (Table 2).

### 2.3. Nutrient Dynamics

To quantify the effects of organic manures under varying input rates on soil nutrients (macro and micro), we quantified total nitrogen (TN), alkali-hydrolyzed nitrogen (AN), total phosphorous (TP), available phosphorous (AP), total potassium (TK), available potassium (AK), DTPA-extractable zinc (Zn), copper (Cu), iron (Fe) and manganese (Mn) at the soil macro- and micro-aggregate fraction level. Manures under different input rate practices notably altered the above-mentioned soil nutrients compared with CK in both investigated soil aggregate fractions in the alpine meadows ecosystem. In addition, the concentration of the above-mentioned nutrients was higher in the macro-aggregate fraction than in the micro-aggregate fraction (Table 3 and Figure 3a–h).

In the soil macro-aggregate fraction, the 8.64 ± 0.31 g kg^−1^ soil TN was higher under sole SM, and 643 ± 16 mg kg^−1^ soil AN was higher under the interaction of sheep manure and cow dung (SMCD) on average across manure input rates than under CK. SM, CD and SMCD increased TN by 16.75, 9.32 and 15.94%, respectively, and AN by 15.36, 20.42 and 25.09%, respectively, over CK across gradients. Almost in parallel with the soil aggregate fraction above, a similar trend was observed in the case of the micro-aggregate fraction, as 8.55 ± 0.30 g kg^−1^ soil TN was higher under SM amendment and 631 ± 13 mg kg^−1^ soil AN was higher under SMCD compared with the control across varying gradients. Minimum TN and AN values were related to CK in both investigated aggregate fractions. Moreover, the too-low as well as too-high input rates of all investigated manures did not improve TN and AN accumulation. The 1.62 kg m^−2^ and 2.16 kg m^−2^ input rates of SM, CD and SMCD depicted positive results in TN and AN improvement (Table 3).

For the soil macro-aggregates, SM achieved the maximum value of 0.93 ± 0.05 g kg^−1^ for soil TP and maximum value of 21.56 ± 0.8 mg kg^−1^ for soil AP across gradients over CK. On average, at various manure input levels, SM, CD and SMCD increased TP by 30.98, 29.57, 28.16%, respectively, and AP by 24.91 and 21.08, 23.40%, respectively, over CK (Table 3). The data demonstrated a similar trend in micro-aggregates, as the highest soil TP value of 0.86 ± 0.05 g kg^−1^ and maximum soil AP value of 20.89 ± 0.7 mg kg^−1^ were associated with sole SM treatment, irrespective of manure input levels. CK had the lowest values for TP and AP for both investigated aggregate fractions. The concentration of TP and AP in the micro-aggregate fraction followed the trend of CK < CD < SMCD < SM across manure input rates. Medium doses of all manures showed higher accumulation of TP and AP for both aggregate fractions of topsoil.

For the macro-aggregate fraction, TK was significantly (*p <* 0.05) (12.42%) higher when SMCD was applied and 7.18% AK was higher when SMCD was applied over CK, regardless of manure gradients. TK and AK followed the trend of CK < CD < SM < SMCD on average over gradients. Regarding TK and AK for micro-aggregates, sole CD treatment produced the maximum 7.19 ± 0.14 g kg^−1^ TK, whilst sole SM produced the 591 ± 15 mg kg^−1^ value for AK over CK across gradients. Too-low levels of all manures failed to increase their improvement for both aggregates under the alpine meadows ecosystem. The lowest values for TK and AK for both aggregate fractions were observed under the control treatment (Table 3).

In the macro-aggregate fraction, manures under varying gradients significantly increased the DTPA-extractable Zn, Cu, Fe and Mn. The highest values of all investigated micronutrients were reached under SMCD treatment, which was followed by sole SM amendment, whilst the lowest values were related to the CK strategy across varying manure gradients (Figure 3a–h). SM, CD and SMCD increased Zn by 11.13, 7.03 and 12.10%, respectively; Cu by 15.30, 12.75 and 17.85%, respectively; Fe by 28.2, 26.7 and 45.10%, respectively; and Mn by 4.63, 3.37 and 6.20%, respectively, compared with the CK average over different manure input rates. Increasing manure input rates resulted in the maximum accumulation of the above-mentioned soil micro-nutrients, with the highest accumulation of soil DTPA-extractable Zn, Cu, Fe and Mn recorded under 2.16 kg m^−2^ and 2.70 kg m^−2^ input rates of applied manures. In the micro-aggregate fraction, maximum concentrations of DTPA micronutrients, i.e., Cu, Fe and Mn, were reached under SMCD, whilst in the case of Zn, sole SM showed the highest value. SM, CD and SMCD increased Zn by 10.20, 5.36 and 9.42%, respectively; Cu by 13.16, 10.38 and 15.14%, respectively; Fe by 24.32, 20.47 and 35.76%, respectively; and Mn by 3.11, 2.38 and 5.50%, respectively, over CK across manure gradients for soil micro-aggregates. Almost in parallel with the macro-aggregate fraction, the increasing manure input rates significantly improved the concentration of the above-mentioned micronutrients. The 2.16 kg m^−2^ and 2.70 kg m^−2^ input rates of applied manures depicted the highest level of micronutrients over the lowest doses of manures (Figure 3a–h).

### 2.4. Forage Biodiversity, Productivity and Nutritional Quality

Analysis of variance results exhibited the great influences of organic manures under varying input rates on the meadow vegetation structure, aboveground biomass and nutritional quality traits (Figure 4a–h). Field investigation data proved that species diversity, evenness, richness and aboveground biomass of alpine meadow community were the highest under all tested manure gradients compared to CK on average. Compared with the control, organic manure treatments SM, CD and SMCD significantly increased the species diversity by 9.67, 6.45 and 4.51%, respectively; species evenness by 1.20, 1.10 and 1.00%, respectively; species richness by 40, 31 and 48%, respectively; and aboveground biomass by 46, 19 and 40%, respectively, regardless of the varying manure input rates (Figure 4a–d). Upon enhancing the manure application rates from 1.62 kg m^−2^, the positive influences on species diversity, evenness, and richness declined. The peak values of these parameters were observed under 1.08 kg m^−2^ and 1.62 kg m^−2^ manure input rates. In the case of alpine meadow community aboveground biomass, the 2.16 kg m^−2^ gradient showed better results compared to other gradients.

The data showed that the forage nutritional quality parameters (CP, CF, CA and WF) in the alpine meadow community were altered to a noteworthy extent with manures SM, CD and SMCD under various levels of input rate application. In the explored manure treatments, the sole sheep manure had significantly (*p <* 0.05) higher CP (102 ± 3.00 g kg^−1^), followed by sole CD amendment, than CK across gradients (Figure 4e). Unlike CP, the CF amongst different manure practices in descending order was noted as sole CD > sole SM, irrespective of applied manure input levels (Figure 4f). Concerning different input rates of manures, 1.62 kg m^−2^ showed better performance in the case of both CP and CF. However, CK had the lowest values for both CP and CF parameters. The information in Figure 4g,h reveals that the CA and WF trended in the opposite direction to CP and CF. The average of manure input rates and the maximum CA and WF values were found under CK treatment compared to SM, CD and SMCD application. The average across gradients of CA followed the trend of CK > CD > SM > SMCD, whilst WF followed the trend of CK > SMCD > CD > SM (Figure 4g–h). Interestingly, considering different gradients, the 2.16 kg m^−2^ of SM had 9.68 ± 0.13%, the highest CA value, whilst the lowest input rate (0.54 kg m^−2^) of SMCD had the maximum WF value in comparison with CK.

### 2.5. PCA and Heatmap Correlation Analysis

Multivariate variation was analyzed under different manure additions for soil characteristics, vegetation biodiversity, aboveground biomass and forage nutritional quality through principle component analysis (PCA), as presented in Figure 5. The PCA analysis allowed for the isolation of five major principal components according to the Jolliffe cut-off value. In PCA analysis, due to the contact of PC1 and PC2, the observation point exhibits the overall variance separately for the five components. The eigenvalues > 1 of PC1 and PC2 were extracted and illuminated 62% of the total variance in PCA analysis of 31 variables (i.e., BD, macro-aggregates, micro-aggregates, SOC, C*frac*_1_, C*frac*_2_, C*frac*_3_, C*frac*_4_, CPI, L, LI, CMI, SI, TN, AN, TP, AP, TK, AK, Zn, Cu, Fe, Mn, diversity, evenness, richness, AGB, CP, CF, CA and WF). However, PC3, PC4, and PC5 did not permit us to add extra information; hence, they are not plotted. The highest PC1 loadings comprise 41.7% of the total variance, whereas the lowest PC2 loadings include 20.3% of the total variance (Figure 5).

A heatmap correlation plot amongst different organic carbon contents, soil properties, vegetation structure, biomass and nutritional quality indicators is shown in Figure 6. Our results revealed that there was a significant positive correlation of carbon contents with carbon management indices, soil macro- and micro-nutrients, vegetation biodiversity, productivity and forage quality attributes. Therefore, the current study displays that a strong correlation exists amid carbon elements, meadow forage parameters and soil indicators. Accordingly, alterations in soil carbon contents and nutrients due to the manure addition amended the biodiversity and, consequently, increased the productivity and forage quality due to essential nutrient supplementation in meadow ecosystems.

## 3. Discussion

This research delivers novel awareness into the alpine meadow ecosystem function of manures under different input rates through their influence on soil BD, aggregates, SOC pools and stabilization as well as nutrients at the aggregate level, vegetation structure, productivity and forage nutritional quality. We revealed that manure addition under varying gradients increased the aggregate size, SOC fractions and stability, which greatly influenced nutrient accumulation, vegetation biodiversity, aboveground biomass and forage nutritional quality.

### 3.1. Effect of Manure Gradient Regimes on Soil BD and Aggregation

BD and aggregation are basic soil physical parameters that remarkably influence SOC sequestration and the availability of essential nutrients in a soil system [53]. Manure addition can markedly decrease BD, increase pore space and aggregate stability and develop overall better soil health characteristics for plants [54]. Our results demonstrated that SM, CD and SMCD under different input rates significantly decreased BD in topsoil over CK (Figure 1a), which concurs with earlier studies [28,55]. Soil porosity development was the dominant influence in all manure gradient amendments, as shown by the noteworthy reduction in soil BD, particularly in SMCD-treated plots. The BD decreased as SM, CD and SMCD input rates increased. However, no significant difference was observed between SM and CD treatments, irrespective of manure input rates, on soil BD. This revealed that the mixture of SM and CD application significantly affected BD, rather than the sole SM and CD application across gradients. The following factors could describe how the manures decreased BD: (i) low-density characteristics and high pore space improved soil total pore volume [54]; (ii) manures increased soil aggregate size, stability [56], macro-pore and macro-aggregate formation induced by the cementing action of polysaccharides [57]; formation of organic acids excreted by microbes during the decomposition of added manures [57]; and (iii) the dilution influence of abundant soil organic matter on soil mineral fractions [58].

The findings of the current study depict that manures with different input rates increased the macro-aggregate fraction compared with CK, which is consistent with the results reported by [29,30]. This may be related to increased soil organic matter under manure addition [59]. Additionally, our results showed that the proportion of soil macro-aggregates increased and was directly proportional to manure input rates. SOM conserves the desired soil physical environment by influencing and enhancing the soil’s physical properties, expressed through void pore space and aggregation [59,60]. Higher soil macro-aggregate fractions with manure addition over CK might be due to a reduction in bulk density under manure application [54]. Moreover, in our study, the concentration of micro-aggregates tended to decline after manure addition. This was due to the soil macro-aggregate fraction formation from the soil micro-aggregates, with the cementing or binding agents from the organic manures [30]. Microbial amino sugar serves as a binding agent, binding primary soil particles to form the soil micro-aggregate fraction, which is then bound and collected to form the soil macro-aggregate fraction [61]. This was similar to the results of Wortmann et al. [62].

### 3.2. SOC Concentration, Fractions and Stabilization Under Manure Management Practices

SOC and SOC fraction dynamics are of critical significance in understanding alpine meadow ecosystem sustainability and health due to the unique climate. This plays a pivotal role in soil structure preservation, soil and water conservation, mitigating CO_2_ emissions, thus promoting soil and environmental health [3,4,25]. Moreover, soil accumulation highly depends upon the relative process rates that add C (for instance, plant photosynthetic activity) and processes that eliminate soil C (such as organic matter respiration and decomposition by soil microbes). Processes that affect the soil C balance are sensitive to anthropogenic drivers; for example, organic and inorganic nutrient fertilization has changed C cycling in several ecosystems [20,21]. In this study, SOC and SOC fractions were more abundant in the macro-aggregate fraction compared with the micro-aggregate fraction under manure input rate addition (Table 1), which agrees with a previous report [63]. Their physical protection by soil aggregates is a vital C stabilization mechanism [64]. Higher SOC concentrations and SOC fractions in macro-aggregates could be attributed to the higher chemical and biophysical protection of the macro-aggregate fraction than in micro aggregates [65]. Moreover, SOC concentrations and SOC fractions such as (C*frac*_1_, C*frac*_2_ and C*frac*_4_) values, except C*frac*_3_ values, in both aggregate fractions increased in our study under manure addition over CK (Table 1). Major factors can clarify how the manures under different input rates increased SOC and SOC fractions: (i) Higher SOC and C*frac*_4_ in manures treatments may be credited to manure addition because manures have a greater proportion of recalcitrant organic compounds [66]. (ii) Meadow productivity was also increased under manure input rates (Figure 4d), which lead to increased total inputs of C from root biomass and rhizodeposition [67]. (iii) The addition of manures can result in enhanced lignin and lignin-like compounds, which are key modules of the resistant SOC pool in the soil system [68]. Maximum SOC and C*frac*_4_ in manure gradient addition (Table 1) showed that manure application can be a useful approach to increase SOC stabilization [69]. (iv) Maximum labile SOC fractions such as C*frac*_1_ and C*frac*_2_ at different input rates of SM, CD and SMCD were due to manure use, because organic manure application increased the microbial activities in soil, thus increasing the conversion of plant residue C into the labile C fractions [70]. (v) Manures have the potential to directly add a labile soil organic C pool because manures are direct and rich organic labile C sources, which were directly added the labile organic C in soils [66]. The decomposition rate is slowed down by the lower temperature at high altitudes, which accounts for increased SOC and SOC fractions. Moreover, the SOC rises in C*frac*_3_ were restricted by manures under input rate application, accordingly signifying that diverse microbial mechanisms occurred for each treatment type. Modifications in both specific extracellular enzyme production and microbe community structures are fundamental factors for the contribution of SOC decomposition in C*frac*_3_ [71], such as *r*-strategy microbes, which are primarily capable of quickly using labile substrates, and low molecular weight would decline under scarce substrates; then, *K*-strategy microbe activity would rise as *K*-strategists effortlessly degrade recalcitrant SOC and bacteriological residues for their nutrients as well as energy [71,72]. Microbes must initially satisfy their nutrient demands, especially nitrogen and energy, to yield extracellular enzymes and degrade communities for successive co-metabolism [72]. Consequently, we speculated that manure addition with different gradients could not provide more energy sources for enzyme production and the idiosyncratic bacterial community. That would increase SOC mineralization at the cost of restricting SOC rises in C*frac*_3_. SM, CD and SMCD with varying input rates displayed a lower SOC rise in the less labile C*frac*_3_. However, the SOC alterations in C*frac*_3_ were not very outstanding, probably due to the soil and climatic conditions as well as the quantity and quality of manure C inputs than an earlier incubation study [72]. Manures with contrasting inorganic nitrogen fertilization might also possibly increase the SOC in the less-labile C*frac*_3_ [73]. Experiments on SOC and SOC fraction dynamics under manure gradients still need more convincing and precise suggestions, for example, a study including soil enzymes and microbes in a long-term experiment. Moreover, the current study indicated that C in C*frac*_1_, C*frac*_2_ and C*frac*_4_ was higher in manure-treated soils over unfertilized control soils, confirming that organic manure inputs drive variations in labile and non-labile C pools. Our findings are supported by many reports [21,22,70], signifying that higher C input practices significantly increase total organic carbon and labile and non-labile C pools in various ecosystems.

The lability index (LI) mostly provides knowledge about SOC fraction types and their oxidization ability [74]. Its greater value suggests more unstable carbon (active C pool), which shows a reduced soil capacity to store C. The carbon management index (CMI) is a convenient indicator to explore the management practices’ capacity to increase soil quality [75]. CMI includes a measure of the effect on total soil carbon (CPI) and the lability of that carbon (LI), therefore reflecting both the potential of nutrient cycling and C sequestration. Meanwhile, the index is the product of these two parameters; only amendments that score highly on CPI and LI will have a greater CMI. Our research demonstrated that the effects of manure input rates on CMI and SI were noteworthy, with CMI decreasing and SI increasing in treatments receiving manure input rates over CK for macro-aggregates. Here, we plan a mechanism in which manure addition with a contrasting inorganic nitrogen management practice may help to regulate SOC dynamics in C*frac*_1_, C*frac*_2_ and C*frac*_3_ and also protect current SOC from further mineralization. Consequently, both manures and inorganic nitrogen will increase the soil CMI for macro-aggregates. However, both CMI and SI were increased under manure addition in the case of micro-aggregates. This is confirmed by several scholars in their long-term fertilization studies [76,77]. The CMI was extremely correlated with the amount of total SOC and each labile and non-labile C fraction. Tirol-Padre and Ladha [78] clarified that CMI differences under fertilizer treatments are credited to the rise in C addition and the organic matter quality variations, thus influencing the C susceptibility to the oxidation of KMnO_4_. Under manure input rates, the rise in SI was due to the rate of labile C loss being higher than that of stable C. This is mostly due to the differential microorganism’s accessibility to diverse active C fractions [79]. The complex structure of stable carbon sheltered it against bacterial attack. Moreover, we found that the SI under manure addition of macro-aggregates was higher than that of micro-aggregates, which might be due to greater SOC’s protection ability [80]. Carbon storage changes related to the macro-aggregate fraction were suggestively affected by manure addition over the micro-aggregate fraction. This specified that the dynamic changes in the C pool are more sensitive in the macro-aggregates compared with micro-aggregates [80]. Therefore, manure input rates improved the soil C pool stability by stimulating microbial active nutrient utilization, altered forage litter quality and macro-aggregate C pools are more sensitive to manure addition.

### 3.3. Influence of Manure Gradient Application on Soil Nutrient Build-Up

Nutrients (macro and micro) play a fundamental role in sustainable and maintainable soil fertility [33]. Also, these nutrients contribute to the productivity development of crops and forage in many ecosystems [31]. Soil nutrients removed as fodder harvest and not substituted by nitrogen fixation or atmospheric deposition and straws must be replenished through organic/inorganic fertilizer inputs; otherwise, soil productivity and fertility will decrease [34]. Moreover, imbalanced fertilization has led to a serious reduction in soil fertility, which threatens the yield of alpine meadows. Manures can pointedly increase soil macro- and micro-nutrients (for instance, N, P, K, Cu, Fe, Zn and Mn) and their bioavailability, increase the physical and biological soil environment, and increase vegetation biodiversity, yields and plant nutritional quality under different climatic conditions [34]. In our study, manure input management practices influenced soil nutrients to diverse extents (Table 3 and Figure 4a–h). A significant improvement was recorded in soil macronutrients (TN, AN, TP. AP. TK and AK) and micronutrients (DTPA-extractable Zn, Cu, Fe and Mn) for both aggregate fractions (micro and macro) explored under 0.54, 1.08, 1.62, 2.16 and 2.70 kg m^−2^ of SM, CD, SMCD practices compared to the control. There are a series of probable direct or indirect mechanisms through which soil nutrients (macro and micro) increase under manure gradient regimes. First, the increase in soil nutrients (TN, TP, TK, AN, AP, AK, DTPA Cu, Fe, Zn and Mn) was due to the organic matter addition through livestock manures [35]. Manures are applied to the soil surface for a shorter time period, resulting in an rise in the concentration of humic acid, denser soil mineral fraction dilution and soil aeration [81]. Second, the minor soil nutrient status in unfertilized CK was credited to the nutrient mining by meadow forage plants without manure addition for 3 years [82]. Third, during the decomposition of manures, the organic acids released hasten the phosphorous solubilisation due to enriched microbial activities and owing to delayed phosphorous fixation in soil that increased TP and AP. Fourth, maximum soil TK and AK under manure gradients may be related to the augmented soil cation exchange capacity, which increased soil capacity to hold more exchangeable K and condensed leaching K losses [83]. Fifth, the maximum concentration of DTPA-extractable micronutrients under manure application might be due to organic manure addition because manures led to metal complex formation with soil organic matter, which enhanced the micronutrient availability by lessening their fixation, precipitation, and adsorption. Sixth, higher micronutrients under manures might be due to nitrogen addition under manures because N addition reduced the pH of soil, which improved the DTPA-extractable micronutrient availability [84]. Differences in soil DTPA-extractable micronutrient levels under manure input rates may be due to soil environment changes, for instance, more organic matter and inferior soil pH. Organic manure not only supplies noteworthy amounts of nutrients but also improves chemical as well as biological reactions that result in soil non-available nutrient dissolution [85]. These findings corroborate the results of previous studies, where the manure addition enhanced the soil nutrient status [81,84]. Moreover, the higher concentration of the above-mentioned soil nutrients in macro-aggregates over micro-aggregates occurred irrespective of manure application under different input rates, which may be owing to the physical protection of soil macro-aggregates.

### 3.4. Forage Traits with Respect to Manure Input Rates

Forage has a noteworthy relationship with animal diet, as it is a vital meat and milk production system in the livestock sector. Therefore, for better forage structure composition, productivity and high nutritional quality, manure application is a proficient technique for meat and milk production in animal husbandry [39,40]. The meadow management strategies herein pointedly differed in terms of their influences on the forage structure, productivity and nutritional quality parameters. These results were consistent with those reported by Socher et al. [40] and Štýbnarová et al. [42]. Our study indicated that the application of minimum rates of sole SM and sole CD treatments significantly increased species diversity, evenness and richness of alpine meadow community compared to high input rates of manures. This showed that short-time manures with the lowest input addition could efficiently restore the alpine meadow vegetation, which may be due to the higher nutrient availability under higher rates of manures and might favor some meadow plant community species that are more responsive to these nutrients, prominent to variations in community plant composition and a decline in meadow species diversity [39]. These results are parallel with Liu et al. [86], who found that higher doses of manure application decreased species diversity, evenness and richness under a grassland ecosystem due to N addition through manures. Nitrogen addition under manure application can reduce the compositional stability of grassland, resulting in a reduction in the evenness and richness of a meadow community (Lai et al. [87]). Moreover, this study showed that sole SM, CD alone and a mixture of SM and CD significantly improved ABG over control, being highest under a 2.16 kg m^−2^ input rate. Manure addition promotes soil water conservation and promotes the growth of the meadow community to a certain extent [40]. These consequences agree with the results of Kacorzyk and Głab [88], who stated that the forage production in alpine grasslands increased under manure amendments. Additionally, the current study confirmed that the use of SM, CD and SMCD under various input applications significantly increased CP and CF over CK. This was owing to the maximum nitrogen content under manure addition, which improved forage nutritional quality by decreasing the C:N ratio [38], in agreement with earlier studies [42,88]. Also, on average, for the different gradients, we found maximum CA and WF values under CK over manure treatments, whilst, interestingly, lower doses of manures increased CA and WF more than CK, which concurs with the study of Simi’c et al. [43].

### 3.5. Correlation

Sheep manure, cow dung and a mixture of both offer the multiple benefits of SOC sequestration and soil physico-chemical properties’ improvement whilst increasing vegetation structure and nutritional quality [1,21,36]. Manure application improves the soil structure, resulting in more wide plant root development, vegetation structure, biomass and improved SOC and nutrient accumulation, micro-flora and fauna activity, which results in better plant nutritional quality [1,38,45]. Therefore, the dynamics of soil characteristics recorded under manure gradients resulted in an amended correlation amongst carbon elements, forage structure parameters with soil indicators. Nevertheless, this research does not comprise a multi-manure setup, which might be considered as an upcoming prospect for study.

### 3.6. Research Limitations

Our study primarily focused on a sole manure treatment and the integration of only two organic manure sources (sheep manure and cow dung) with varying gradients for the short term rather than multi-manure sources (poultry manure, sheep manure, pig manure and cow dung) under varying input rates for the long term. Nevertheless, future studies are needed to examine the influences of long-term multi-manure treatments with contrasting inorganic nitrogen applications on soil organic carbon stability, soil and forage quality. Moreover, studies are needed on more labile SOC fractions, i.e., microbial biomass C, particulate organic C, dissolved organic C, light fraction organic C, soil and forage quality indicators and global warming potential under long-term multi-manure practice to produce more powerful data for alpine meadow management/restoration and environmental sustainability in Qinghai–Tibetan Plateau ecosystems.

## 4. Materials and Methods

### 4.1. Field Site Description

Our current scientific research was conducted in Tianzhu Tibetan autonomous canton (37°10′ N, 102°47′ E; 2940–3500 m a.s.l.) of Gansu province under the supervision of Gansu Agricultural University, which is positioned at Northeastern fringe of Qinghai–Tibet Plateau in China (Figure 7). Tianzhu has typical plateau climatic conditions categorized by thin air, dry, cold, strong solar radiation and low barometric pressure. Long-term average annual rainfall was 416 mm (75% rainfall concentrated between June and September whilst long-term average annual temperature was 0.16 °C (11.2 °C in July and −11.4 °C in January)). Average 55 y annual evaporation is 1592 mm, while mean annual sunshine is 2600 h, with approximately 120 days of plant growth stage. The study site has alpine chernozem soil type with 7–15% sand contents (2–0.05 mm), 66–74% silt contents (0.05–0.002 mm) and 15–20% clay contents (<0.002 mm). In accordance with Chinese soil classification, it is categorized as an alpine steppe soil [89], which corresponds to a typical Cryrendoll in accordance with classification of USDA [90]. Multiple grassland types are present in Tianzhu Tibetan Plateau, but alpine meadow, alpine shrub, shrubland–grassland ecotone and herbs are dominant. In the research region, the alpine meadow dominated by *Kobresia humilis*, *Elymus nutans* and *Stipa aliena*, with >80% meadow coverage is scattered under the altitude of 3000 m above sea level. The dominant species in the alpine shrubland are *Rhododendron thymifolium*, *Salix oritrepha* and *Rhododendron capitatum* with >40% shrub coverage at above an altitude of 3020 m above sea level. Shrubland–grassland ecotone dominated by *Potentilla fruticose* with 5–40% shrub coverage is scattered at altitude of 3000–3020 m above sea level. The dominant herb species are *Polygonum polygonum* (*Polygonum viviparum*) and *Equisetum arvense*, with approximately 60–85% coverage at research site at altitude of 2960–3020 m above sea level. Before 1984, there were no cattle fences in Tianzhu belt, and whole grassland region was a public pasture. Local herdsmen said in interviews, traditionally, grassland in the research area was grazed by sheep (*Ovis aries*) and yak (*Bos grunniens*). Livestock quantity, species, and grazing period were identical all over the area, and the vegetation was comparatively steady. Meanwhile, the Household Responsibility System was employed in early 1984; each household herdsman managed grassland area, and cattle fences were built in Tianzhu. Livestock periods and amount of grazing have altered because of diverse grazing management practices of individual herdsmen on the grassland. The soil and forage quality and productivity were reduced in the alpine meadow, alpine shrubland and alpine shrubland–grassland ecotone due to influence of livestock over grazing in the long run. For that reason, all grassland types, including alpine meadow, alpine shrubland and shrubland–grassland ecotone, experienced severe degradation by 2014. In accordance with the alpine grassland degradation criteria (based on soil properties and vegetation), the alpine meadow belongs to the category of light degraded [91].

### 4.2. Research Design and Treatment Details

This study was carried out as part of ongoing research originally set up in 2022 with manure addition under different application times (March and September), which was continued in successive years during 2023–2024. The results of only manure gradient application in March are presented in this study. The alpine meadow field had been subjected to conventional livestock grazing by yak and Tibetan sheep prior to manure application. Cattle fences were built across the entire alpine meadow study area to protect it from livestock grazing and human disturbance. The present study of manure application under different gradients was conducted using a randomized complete block design (RCBD) with four replications. The research comprised four treatments, counting one un-amended (CK) control and three diverse manure practices, specifically, sheep manure alone (SM), sole cow dung (CD) and mixture of 50% sheep manure and 50% cow dung (SMCD) with five gradients, i.e., 0.54, 1.08, 1.62, 2.16 and 2.70 kg m^−2^, using 7.50, 15, 22.50, 30, 37.50 g m^−2^ nitrogen rates. All experimental treatments were implemented in 64 plots, including four replicates of each treatment. Each individual plot covered an area of 30 m^2^ (6 m × 5 m), and a 1 m buffer was set between two adjacent plots without manure application. These 64 plots are protected from animal grazing and human disturbance by fencing systems. Experimental materials SM and CD were taken from the nearby local farm of livestock manure that was rotting for more than six months. The nutrient status of SM and CD was as follows: total nitrogen (TN) of 0.16 ± 11 g kg^−1^ and 0.15 ± 14 g kg^−1^, respectively, total phosphorous (TP) 0.12 ± 2.8 g kg^−1^ and 0.13 ± 3.65 g kg^−1^, respectively, and total potassium (TK) 0.11 ± 15 g kg^−1^ and 0.08 ± 18.40 g kg^−1^, respectively. Typical broadcast approach was followed to apply different manures uniformly on the soil surface of each plot in March of every year. The implemented manures with five different gradients and gradient average (GM) values were compared with the control. The management practices were ensured during the vegetation-growing season every year.

### 4.3. Properties of Alpine Meadows Soil

Before different manure applications, random soil sampling was conducted from 8 different positions, then blended for a representative soil sample. Basic characteristics of alpine meadow soil at 0–10 cm soil depth are shown in Table 4. Alpine meadow soil contained average soil organic carbon (SOC) of 3.20 ± 78.3 g kg^−1^, total nitrogen (TN) 0.50 ± 7.12 g kg^−1^, total phosphorous (TP) 0.04 ± 0.70 g kg^−1^, total potassium (TK) 0.33 ± 6.55 g kg^−1^, pH 0.22 ± 7.8, bulk density (BD) 0.03 ± 0.78 g cm^−3^ and 0.40 ± 70.8% pore space.

### 4.4. Soil Sampling and Sample Preparation

Soil samples were collected from 64 plots in last week of August during 2023–2024 with an auger (diameter: 8 cm; length: 5 cm), in order to evaluate the influence of manure gradients on soil aggregates, SOC concentration and SOC fractions and nutrients. From each plot, the soil sampling was performed at top surface layer 0–10 cm soil depth. Five soil samples from different treatments, including four replicates, were placed in plastic bags and transported to Gansu Agricultural University for chemical analysis. Moist meadow field-augured soil samples were softly broken up with natural weak planes, and observable plant residues were manually removed using tweezers. Augured soil samples were air-dried for 6–8 days and sieved with 2 mm sieve so as to separate the fine soil fraction (fine soil material having <2 mm diameter) and the coarse fraction (stones and gravel having diameter of >2 mm). Coarse soil fraction was demolished during this stage. Then, air-dried soil samples were passed through sieves with dimensions of 2 mm and 0.25 mm to determine weight by particle size, and then we sieved soil macro-aggregates, and the micro-aggregate fraction samples were stored at 4 °C for further analysis.

### 4.5. Soil Measurements

#### 4.5.1. Soil Bulk Density and Aggregate Measurement

Soil bulk density determination was completed from undisturbed core samples. Five soil samples were taken from each individual plot involving 4 replicates. The soil samples were collected with a core sampler (steel cylinders of 4 cm diameter and 3 cm in length) for BD determination. Sampled soil samples were processed in accordance with procedure described by Bao [94]. The BD was calculated using Equation (1):BD = M/V(1)
where BD = soil bulk density of soil (g cm^−3^); M = dry soil sample mass (gm); V = sample volume (cm^−3^).

We sampled five disturbed soil samples from different treatments at the end of growing seasons of alpine meadow vegetation every year for soil aggregate analysis. The determination of soil mechanical stable aggregate classification was conducted based on dry-sieving approach [95]. In brief, the 200 g air-dried soil sample was weighed and placed on sieves in top layer of set with dimensions of 2 mm and 0.25 mm to determine weight by particle size. Mechanically stable aggregate sieve set was gently shaken manually at a speed of once per second for nearly 40 s for about 5 min to achieve the aggregate size grading. Aggregates with particle size > 0.25 mm are named macro-aggregates, whilst aggregates with particle size < 0.25 mm are named micro-aggregates.

#### 4.5.2. Soil Organic Carbon, SOC Fractions and Nutrient Analysis

The SOC and SOC fractions were analyzed from five soil samples of different treatments for macro- and micro-aggregate fractions. On the basis of K_2_Cr_2_O_7_–H_2_SO_4_ wet oxidation method [92], we determined SOC content. Permanganate-oxidizable SOC fractions are more sensitive to variations in climatic conditions and soil management techniques than SOC, particulate organic carbon and microbial biomass carbon, hence making it a rapid and cost-effective way to assess differences in labile SOC pools [13,14]. We used oxidizing solution gradients, 33, 167, and 333 mmol L^−1^ KMnO_4_, for permanganate-oxidizable soil organic C fraction determination [96]. Briefly, we mixed 0.5 g air-dried soil with a 25 mL KMnO4 solution (33, 167, 333 mmol/L), then shook and centrifuged it. We extracted the supernatant and diluted it with deionized water in a 1:250 ratio; we then read the absorbance of sample at 565 mm with a spectrophotometer and calculated the oxidized SOC content according to the consumption of KMnO_4_. Then, the SOC was separated into four fractions with different degrees of lability:SOC part oxidized under 33 mmol L^−1^ KMnO_4_ denoted as very-labile fraction (C*frac*_1_).Extra SOC part oxidized under 167 mmol L^−1^ KMnO4 defined as labile fraction (C*frac*_2_).Extra SOC part oxidized under 333 mmol L^−1^ KMnO_4_ signified as less-labile fraction (C*frac*_3_).SOC not oxidized under 333 mmol L^−1^ KMnO4 mentioned as non-labile/recalcitrant fraction (C*frac*_4_) [97].

Carbon management index (CMI) [98] was calculated by using Equations (2)–(5) and soil carbon stability index (SI) [99] derivation was calculated by using Equation (6):CPI = Sample soil SOC/Reference soil SOC(2)Lability (L) = KMnO_4_-C/non-labile C(3)LI = Lability of C in sample soil/Lability of C in reference soil(4)CMI = Carbon pool index (CPI) × Lability index (LI) × 100(5)SI = C*frac*_4_ / C*frac*_1_ + C*frac*_2_ + C*frac*_3_(6)

The non-labile carbon is determined as the difference between total soil organic carbon (SOC) content and labile carbon (KMnO_4_-C) content of soil. In this study, the soil sampled in CK is taken as the reference soil.

Nutrient (macro and micro) analysis was performed from mechanically stable soil macro- and micro-aggregate fractions. The concentration of soil total nitrogen (TN) was measured using the Kjeldahl method; available nitrogen (AN) was measured using the method of alkaline hydrolysis diffusion; soil total phosphorus (TP) was measured with an ammonium molybdate colorimetric method; soil-available phosphorus (AP) was determined using sodium bicarbonate extraction and a molybdenum antimony colorimetric method; soil total potassium (TK) was analyzed using the alkali fusion-flame photometric method; available potassium (AK) was determined using extraction with ammonium acetate and then analyzed with flame photometry [93]. The concentrations of DTPA-extractable zinc (Zn), copper (Cu), iron (Fe) and manganese (Mn) were determined using an Atomic Absorption Spectrometer (Varian SPECTRAA 2207 model, Varian Associates, Palo Alto, CA, USA) [94]. The details of the parameters for determination of above-mentioned micronutrients in SPECTRAA2207 are shown in Table 5.

### 4.6. Forage Traits Analysis

In this research, the plant data were measured in August of 2024 during peak of vegetation growth. We established sampling quadrates (1 m × 1 m) per treatment, and in order to elucidate the edge influence, the individual quadrate location was set at least 1 m from the margin. Total coverage in each quadrat was recorded, and the coverage, abundance, height, density of plant species and aboveground biomass in the quadrat were measured. The aboveground biomass (AGB) of green plants of the respective species was determined by ground cutting and placed in marked paper bags. AGB of each species was weighed after drying at 65 °C for 48 h. Shannon–Wiener diversity index, richness index and evenness index were calculated based on Equations (7)–(10):(7)$$H=-\sum_{\left\{i=0\right\}}^{\left\{s\right\}}\left.\left(P_{i}\times lnP_{i}\right.\right)$$(8)$$Pi=\frac{n_{i}}{N}$$(9)R=S(10)$$J=\frac{n_{i}}{N}$$
where *S* is the total number of species in the quadrat, *Pi* is the relative abundance of species *i, n_i_* is the total individual number of the ith species, and *N* is the total individual number in the meadow community.

The determination of crude protein (CP) content, crude fiber (CF) content, crude ash (CA) content and washing fibers (WF) was completed by following the standard procedures of Yuan et al. [100].

### 4.7. Statistical Analysis of Experimental Data

All preliminary data were pretreated in Office Excel 2019 (Microsoft Corp., Redmond, WA, USA), and then one-way analysis of variance (ANOVA) appropriate for randomized complete block design (RCBD) using Statistical Package for Social Science (SPSS) software program on computer for window version 26.0 (SPSS Inc., Chicago, IL, USA, 2007) was performed. To identify significant differences (*p* < 0.05) between manure gradient practices in terms of their soil quality indicators and forage quality attributes, a post hoc test (Duncan’s test) was the procedure. The data of different treatments are shown as mean of four replications ± standard error. Computer software program Origin 2022 (Origin Lab Corporation, Northampton, MA, USA) was used to draw figures in this study. Principle component analysis (PCA) was used to detect multivariate variability introduced by the manure gradient treatments for soil properties and forage quality indicators. Based on the influences of manure gradients on soil characteristics and forage traits, a Pearson heatmap correlation analysis was performed in order to describe the relationship between the investigated parameters.

## 5. Conclusions

Alpine meadows are both the source and victim of a changing environment. Whilst environmental changes exert adverse influences on meadow production and livestock livelihoods in animal husbandry, undesirable meadow restoration and management practices would hasten the climate change process through CO_2_ emissions. China aims to decrease annual CO_2_ emissions arising from grassland industry through the progress of a climate-smart meadows/grassland investment plan. Managing soil carbon pools in grasslands through organic manure management is a significant way to respond to China’s “double carbon” target and a vital measure to maintain the sustainable development of alpine meadow ecosystems on the Qinghai–Tibet Plateau. This experiment concludes that sole sheep manure (SM), cow dung alone (CD) and the integration of 50% sheep manure and 50% cow dung (SMCD) management practices significantly reduce soil bulk density, micro-aggregates, less-labile carbon fraction and forage washing fibers in alpine meadows under a 2.16 kg m^−2^ input rate. In addition, SM, CD and SMCD under 2.16 kg m^−2^ gradient increased the soil aggregate-associated carbon storage, carbon stability and soil nutrient status, forage biodiversity, aboveground biomass and forage nutritional quality attributes in the alpine meadow, as compared to the unfertilized control. Moreover, compared with micro-aggregates, the macro-aggregate-associated carbon pool was more sensitive to manure addition and displayed a greater variation. Furthermore, higher nutrient accumulation was found in peds over micro-aggregates. Hence, our research concludes that the application of sole SM under a 2.16 kg m^−2^ input rate regime leads to an enhancement of carbon stabilization, nutrient accumulation, meadow productivity and forage quality. Accordingly, sole SM under a 2.16 kg m^−2^ input rate practice should be recommended and promoted amongst the household herdsmen in alpine meadows to increase soil and water conservation, soil and environmental quality, and sustainability. In the future, more studies related to higher soil organic carbon fractions and genes involved in carbon sequestration under long-term multi-manure application are necessary. Therefore, this research highlights the need for sustainable meadow management that can facilitate a higher sequestration of carbon, which eventually increases the soil quality, vegetation traits and ecological balance, and, ultimately, serve the global sustainability purpose. The addition of organic manures is more efficient in increasing carbon stabilization, physico-chemical properties and vegetation traits. In essence, our research significantly contributes to understanding the effect of organic manures under varying input rates on soil organic carbon stability, soil and forage quality as well as productivity, offering valuable insights for mitigating climate change and overgrazing and managing soil, environmental and vegetation quality in alpine meadow ecosystems. As such, for sustaining alpine meadow sustainability in Tianzhu, China, SM under a 2.16 kg m^−2^ input rate is a forward-thinking and ecologically sound practice. This research will also be supportive for grassland use policymakers to strategically plan suitable management/restoration techniques for ecological grassland use management in extremely frangible alpine meadow ecosystems and other analogous places in the world.

## Figures and Tables

**Figure 1 plants-14-01442-f001:**
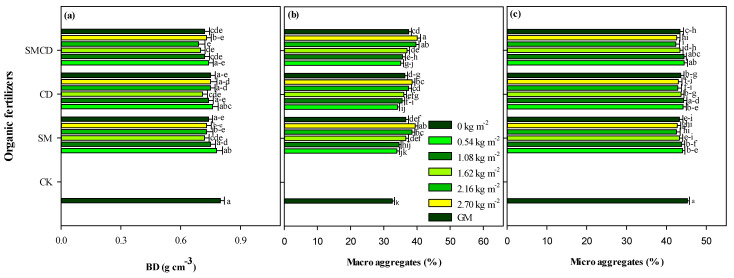
Effect of organic manure gradient practices on soil bulk density and aggregation under alpine meadows of Qinghai–Tibetan Plateau ecosystem. (**a**) The soil bulk density influenced by manure gradient practices; (**b**) the soil macro-aggregate fraction under manure gradients; and (**c**) the soil micro-aggregate fraction affected by organic fertilization. The horizontal error bars represent the corresponding standard error of treatment means; *n* = 4. Small letters denote significant differences between different gradients of manures and average of gradients at *p <* 0.05 (Duncan’s test performed for mean separation). CK: unfertilized control; SM: sheep manure; CD: cow dung; SMCD: mixture of sheep manure and cow dung; GM: gradient average.

**Figure 2 plants-14-01442-f002:**
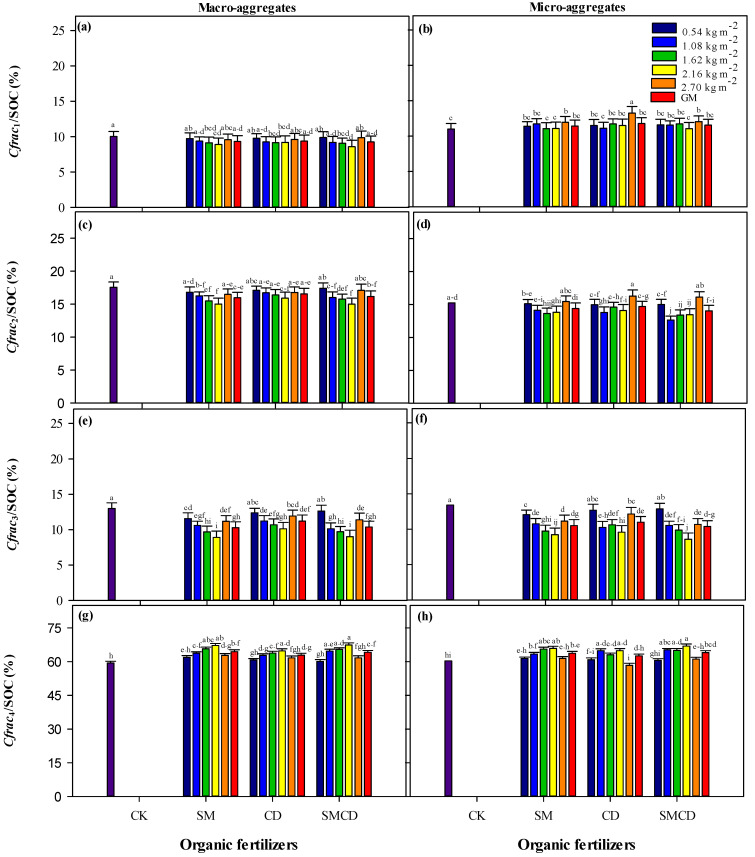
Permanganate-oxidizable SOC fractions C*frac*_1_, C*frac*_2_, C*frac*_3_ and C*frac*_4_ as a proportion of total SOC (%) in different manure input rates. CK: unfertilized control; SM: sheep manure; CD: cow dung; SMCD: mixture of sheep manure and cow dung. Vertical error bars signify the standard error of mean values. Lowercase letters in the same soil aggregate fraction display significant differences amongst different gradients of manures and average of different gradients at *p <* 0.05 (Duncan’s test performed for mean separation). Note: (**a**–**h**) are the C*frac*_1_, C*frac*_2_, C*frac*_3_ and C*frac*_4_ values as a proportion of total SOC under varying manure gradients.

**Figure 3 plants-14-01442-f003:**
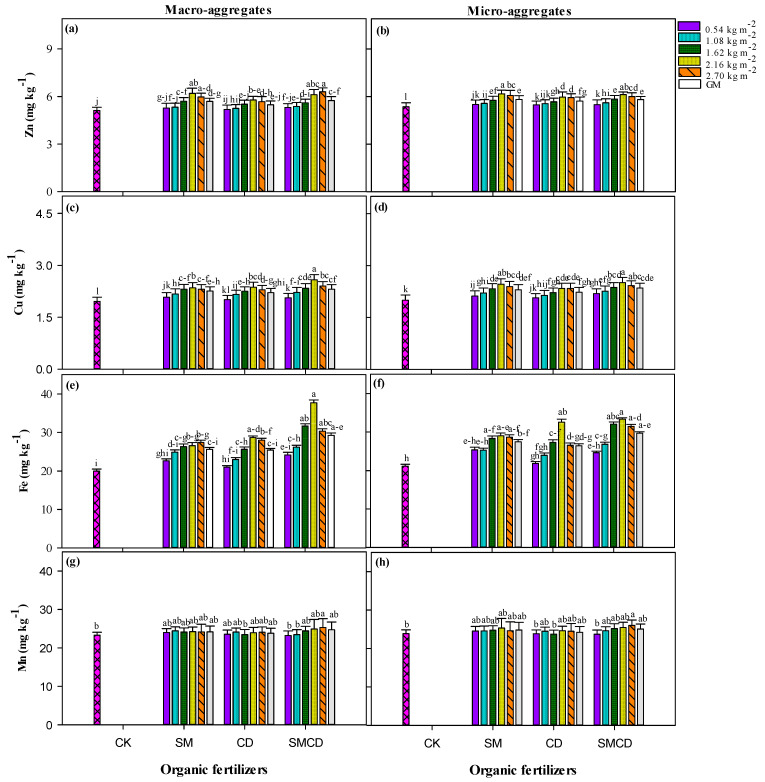
Influence of manure gradient regimes on soil aggregate-associated micronutrients. CK: unfertilized control; SM: sheep manure; CD: cow dung; SMCD: mixture of sheep manure and cow dung. Vertical error bars represent the standard error of means. Lowercase letters in the same soil aggregate fraction display significant differences amongst different gradients of manures and average of different gradients at *p <* 0.05 (Duncan’s test performed for mean separation). Note: (**a–h**) are the zinc, copper, iron and manganese values under different manure input rates on grassland ecosystem.

**Figure 4 plants-14-01442-f004:**
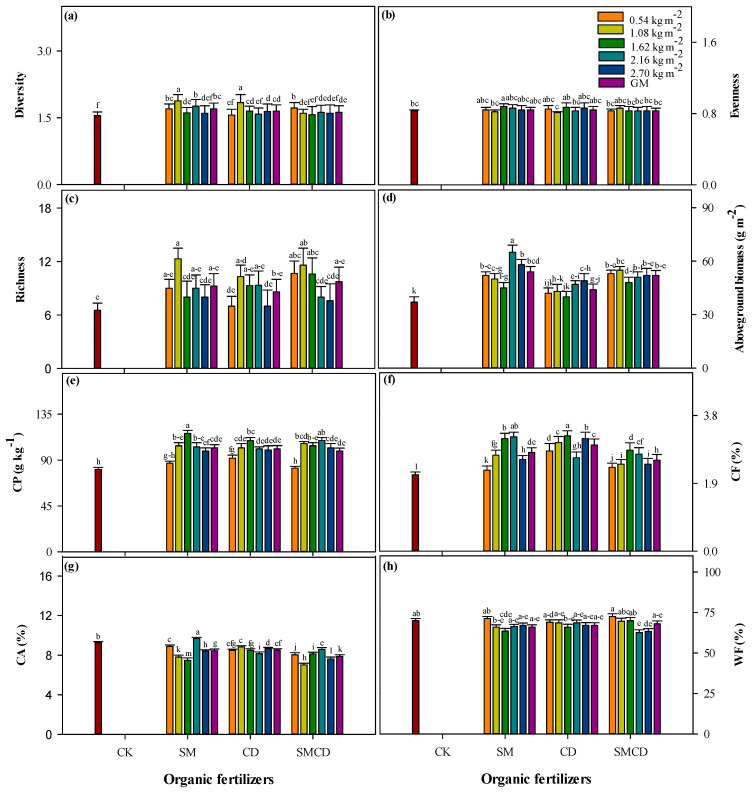
Vegetation structure, productivity and forage nutritional quality traits as influenced by manure input rates. CK: unfertilized control; SM: sheep manure; CD: cow dung; SMCD: mixture of sheep manure and cow dung. Vertical error bars represent the standard error of means. Lowercase letters display significant differences between different gradients of manures and average value of different manure input rates at *p <* 0.05 (Duncan’s test performed for mean separation). Note: (**a**–**h**) are the diversity, evenness, richness, aboveground biomass, crude protein, crude fat, crude ash content and washing fiber values under varying treatments.

**Figure 5 plants-14-01442-f005:**
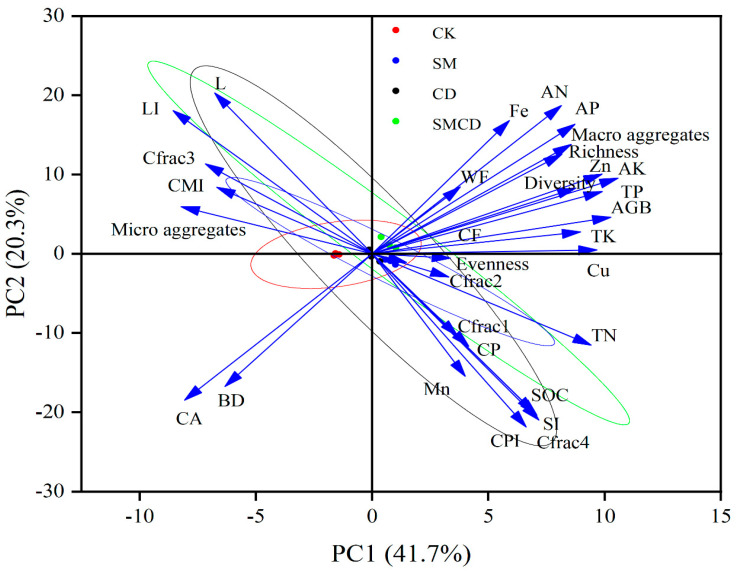
Principal component analysis of soil properties, carbon contents, biodiversity, productivity and vegetation nutritional quality under alpine meadows ecosystem.

**Figure 6 plants-14-01442-f006:**
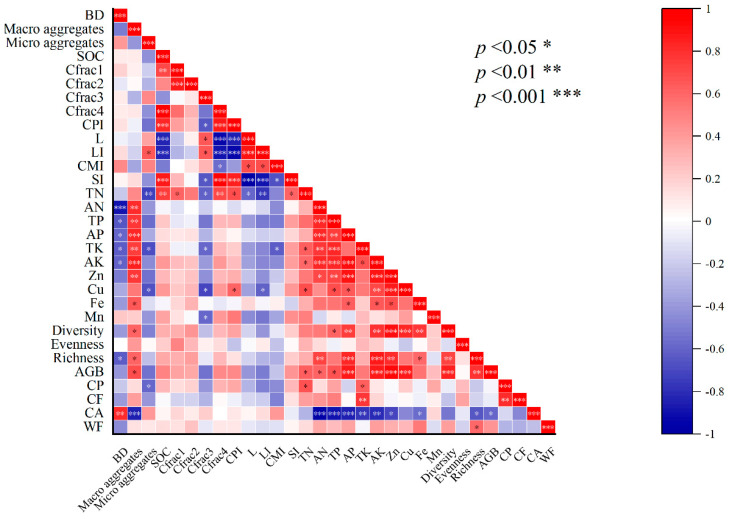
Heatmap correlation of organic carbon contents, soil properties and vegetation structure, productivity and forage nutritional quality indicators. Indicates significance at: * *p* < 0.05, ** *p* < 0.010, and *** *p* < 0.0010. Note: the abbreviated words stand for BD = soil bulk density; SOC = soil organic carbon; C*frac*_1_ = very-labile C fraction; C*frac*_2_ = labile C fraction; C*frac*_3_ = less-labile C fraction; C*frac*_4_ = non-labile C fraction; CPI = carbon pool index; L = carbon lability; LI = carbon lability index; CMI = carbon management index; SI = carbon stability index; TN = total nitrogen; AN = available nitrogen; TP = total phosphorous; AP = available phosphorous; TK = total potassium; AK = available potassium; Zn = zinc; Cu = copper; Fe = iron; Mn = manganese; AGB = aboveground biomass; CP = crude protein; CF = crude fat; CA = crude ash content; WF = washing fibers.

**Figure 7 plants-14-01442-f007:**
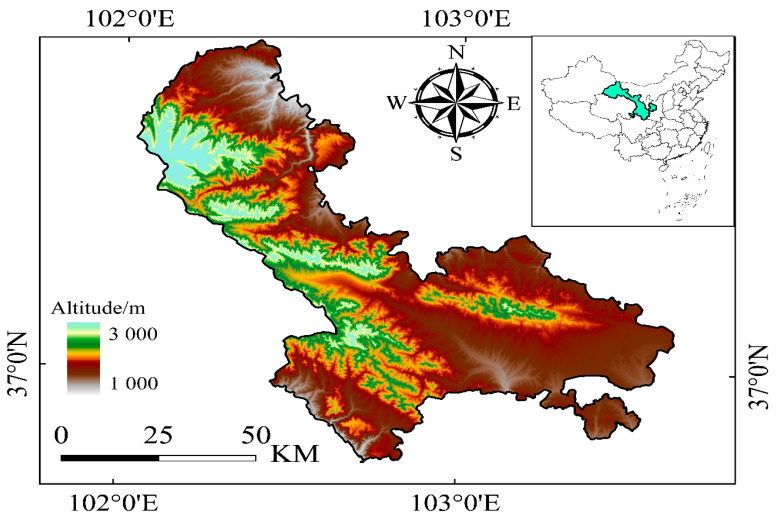
Map of the research zone in Tianzhu of Gansu province, China. Apply ArcGIS 10.2 Software production. Basic geographic information data obtained from resource and environmental science and data center (http://www.resdc.cn/, accessed on 5 February 2025).

**Table 1 plants-14-01442-t001:** Influences of manure input rates on soil organic carbon and permanganate-oxidizable organic carbon fractions.

Fertilization Gradients (kg m^−2^)	Treatments														
Soil Aggregate Fraction	SM	CD	SMCD	SM	CD	SMCD	SM	CD	SMCD	SM	CD	SMCD	SM	CD	SMCD
Macro-aggregates	SOC (g kg ^−1^)			C*frac*_1_ (g kg ^−1^)			C*frac*_2_ (g kg ^−1^)			C*frac*_3_ (g kg ^−1^)			C*frac*_4_ (g kg ^−1^)		
CK	88 j ± 2.6	-	-	8.83 f ± 0.2	-	-	15.46 g ± 0.5	-	-	11.45 a ± 0.4	-	-	52.31 h ± 1.1	-	-
0.54	96.3 ghi ± 1.3	90 ij ± 2.4	89.5 j ± 1.2	9.37 cde ± 0.3	8.86 f ± 0.2	8.84 f ± 0.1	16.18 de ± 0.4	15.50 g ± 0.5	15.57 fg ± 0.8	11.15 abc ± 0.5	11.22 ab ± 0.4	11.30 ab ± 0.6	59.61 d–g ± 1.2	55.04 fgh ± 1.8	53.85 gh ± 1.9
1.08	101 d–g ± 2.5	98 fgh ± 3.7	104 cde ± 2.8	9.50 b–e ± 0.2	9.08 ef ± 0.4	9.62 bcd ± 0.3	16.47 b–e ± 0.6	16.39 cde ± 0.4	16.76 a–d ± 0.7	10.74 d–i ± 0.6	11.01 b–e ± 0.5	10.59 g–j ± 0.6	64.62 bcd ± 2.3	61.51 c–f ± 3.2	67.68 bc ± 2.4
1.62	108 abc ± 3.4	102 c–g ± 2.8	107 bcd ± 1.9	9.86 ab ± 0.4	9.23 def ± 0.3	9.69 abc ± 0.5	16.73 a–d ± 0.7	16.55 a–e ± 0.9	16.90 abc ± 0.7	10.46 h–k ± 0.7	10.80 c–h ± 0.8	10.39 ijk ± 0.7	70.94 ab ± 3.1	64.41 bcd ± 2.7	70.01 ab ± 2.6
2.16	113 a ± 2.2	104.2 cde ± 3.2	112 ab ± 1.6	10.10 a ± 0.5	9.61 bcd ± 0.4	9.76 abc ± 0.4	17.06 ab ± 0.9	16.67 a–e ± 0.6	17.08 a ± 0.9	10.10 k ± 0.8	10.60 f–j ± 0.9	10.24 jk ± 0.9	76.27 a ± 2.6	67.84 bc ± 2.9	76.69 a ± 2.7
2.70	98.8 e–h ± 3.6	92.7 hij ± 1.6	96.4 ghi ± 1.5	9.45 b–e ± 0.3	8.90 f ± 0.2	9.49 b–e ± 0.10	16.30 de ± 0.8	15.56 fg ± 0.6	16.46 b–e ± 0.4	11.05 bcd ± 0.6	11.07 bcd ± 0.4	10.97 b–f ± 0.9	62.03 cde ± 2.7	57.19 e–h ± 3.3	59.38 d–g ± 1.4
GM	103 c–f ± 2.6	97.6 fgh ± 2.7	102 c–g ± 1.8	9.65 bcd ± 0.34	9.13 ef ± 0.3	9.47 b–e ± 0.28	16.54 a–e ± 0.68	16.13 ef ± 0.6	16.55 a–e ± 0.7	10.65 e–i ± 0.64	10.94 b–g ± 0.6	10.61 f–j ± 0.7	66.75 bc ± 2.4	61.21 c–f ± 2.8	65.61 bcd ± 2.2
**Micro-aggregates**															
CK	82 j ± 1.1	-	-	9.00 m ± 0.2	-	-	12.5 k ± 0.4	-	-	11 bc ± 0.2	-	-	49.50 k ± 1.4	-	-
0.54	90 hi ± 1.5	88.2 ij ± 2.1	86.3 ij ± 1.3	10.25 k ± 0.9	10.15 kl ± 0.4	10.00 l ± 0.2	13.60 i ± 0.6	13.20 j ± 0.9	12.95 j ± 0.8	10.86 cd ± 0.5	11.21 a ± 0.4	11.11 ab ± 0.5	55.27 h–k ± 1.9	53.70 ijk ± 2.1	52.26 jk ± 3.2
1.08	97.7 d–g ± 2.3	101 c–f ± 2.6	103 bcd ± 2.1	11.45 fgh ± 0.8	11.22 ij ± 0.6	11.89 cd ± 0.5	13.81 hi ± 0.5	13.93 gh ± 0.7	13.03 j ± 0.6	10.53 e ± 0.6	10.37 fgh ± 0.5	10.87 cd ± 0.6	61.94 d–g ± 2.8	65.48 b–e ± 3.6	67.21 bcd ± 4.3
1.62	105 bc ± 1.8	97 d–g ± 1.4	104 bcd ± 2.7	11.60 ef ± 0.8	11.36 ghi ± 0.5	12.11 b ± 0.4	14.31 def ± 0.6	14.15 efg ± 0.8	13.83 hi ± 0.5	10.21 ij ± 0.6	10.30 hij ± 0.5	10.19 jk ± 0.7	68.86 bc ± 3.3	61.17 d–h ± 3.5	67.19 bcd ± 3.6
2.16	109 ab ± 2.5	102 cde ± 1.7	113 a ± 2.9	12.04 bc ± 0.7	11.76 de ± 0.8	12.54 a ± 0.9	15.07 ab ± 0.7	14.40 de ± 0.8	15.30 a ± 0.8	10.05 kl ± 0.5	9.81 mn ± 0.4	9.73 n ± 0.8	71.83 ab ± 4.2	66.36 bcd ± 4.7	76.09 a ± 3.9
2.70	96.1 e–h ± 3.6	89.7 hi ± 2.8	92.6 ghi ± 3.2	11.48 fgh ± 0.4	11.87 d ± 0.9	11.14 j ± 0.6	14.85 bc ± 0.8	14.57 cd ± 0.9	14.92 b ± 0.9	10.72 d ± 0.4	10.89 c ± 0.7	9.90 lm ± 0.9	59.04 f–i ± 3.4	52.36 jk ± 4.4	56.69 g–j ± 4.7
GM	99.5 c–f ± 2.3	95.4 fgh ± 2.1	100 c–f ± 2.4	11.36 ghi ± 0.7	11.27 ij ± 0.6	11.53 fg ± 0.5	14.32 def ± 0.6	14.04 fgh ± 0.8	14.00 gh ± 0.7	10.47 efg ± 0.5	10.51 ef ± 0.5	10.35 ghi ± 0.7	63.40 c–f ± 3.1	59.83 e–i ± 3.6	63.90 c–f ± 3.9

Data represent means ± SE (*n* = 4). Means with different lowercase letters in the same aggregate fraction indicate significant differences amongst different treatments at *p* < 0.05 (Duncan’s multiple range test was performed for mean separation). Abbreviated words stand for SOC, soil organic carbon; C*frac*_1_, very-labile SOC fraction; C*frac*_2_, labile SOC fraction; C*frac*_3_, less-labile SOC fraction; C*frac*_4_, non-labile SOC fraction. CK: unfertilized control; SM: sheep manure; CD: cow dung; SMCD: mixture of sheep manure and cow dung; GM, gradient average.

**Table 2 plants-14-01442-t002:** Soil carbon pool index (CPI), lability (L), lability index (LI), carbon management index (CMI) and stability index (SI) under different treatments.

Fertilization Gradients (kg m^−2^)	Treatments														
Soil Aggregate Fraction	SM	CD	SMCD	SM	CD	SMCD	SM	CD	SMCD	SM	CD	SMCD	SM	CD	SMCD
Macro-aggregates	CPI			L			LI			CMI			SI		
CK	1.00 i ± 0.00	-	-	0.68 a ± 0.02	-	-	1.00 a ± 0.00	-	-	100 a ± 0.00	-	-	1.46 g ± 0.04	-	-
0.54	1.09 fgh ± 0.04	1.02 hi ± 0.03	1.01 i ± 0.02	0.61 a–d ± 0.02	0.64 ab ± 0.02	0.66 ab ± 0.02	0.90 a–d ± 0.04	0.94 ab ± 0.02	0.97 ab ± 0.03	98.58 a ± 4.30	97.39 ab ± 3.55	98.77 a ± 6.45	1.62 d–g ± 0.06	1.54 fg ± 0.04	1.50 fg ± 0.08
1.08	1.15 c–f ± 0.02	1.11 efg ± 0.02	1.18 bcd ± 0.05	0.56 cde ± 0.02	0.59 b–e ± 0.03	0.54 c–f ± 0.02	0.83 cde ± 0.05	0.86 b–e ± 0.03	0.79 d–g ± 0.03	95.70 ab ± 8.20	96.59 ab ± 7.23	95.05 ab ± 7.12	1.76 b–e ± 0.05	1.69 c–f ± 0.06	1.82 bc ± 0.07
1.62	1.22 ab ± 0.05	1.14 c–f ± 0.03	1.21 bc ± 0.07	0.52 ef ± 0.03	0.56 cde ± 0.02	0.51 ef ± 0.03	0.76 efg ± 0.03	0.84 cde ± 0.03	0.77 efg ± 0.04	93.76 ab ± 6.90	95.30 ab ± 7.66	93.93 ab ± 5.97	1.91 ab ± 0.04	1.77 b–e ± 0.07	1.89 ab ± 0.09
2.16	1.28 a ± 0.06	1.19 bcd ± 0.04	1.29 a ± 0.06	0.48 f ± 0.02	0.54 def ± 0.03	0.47 f ± 0.02	0.71 fg ± 0.04	0.79 d–g ± 0.04	0.70 g ± 0.04	92.19 b ± 9.40	94.62 ab ± 8.88	91.39 b ± 8.76	2.04 a ± 0.07	1.83 bc ± 0.06	2.06 a ± 0.08
2.70	1.12 d–g ± 0.05	1.05 ghi ± 0.04	1.09 fgh ± 0.06	0.59 b–e ± 0.02	0.62 abc ± 0.03	0.63 abc ± 0.03	0.87 b–e ± 0.04	0.91 abc ± 0.04	0.92 abc ± 0.06	97.44 ab ± 9.80	95.76 ab ± 7.44	99.57 a ± 8.81	1.68 c–f ± 0.08	1.61 efg ± 0.04	1.60 efg ± 0.09
GM	1.17 b–e ± 0.04	1.10 efg ± 0.03	1.16 b–f ± 0.05	0.55 c–f ± 0.02	0.59 b–e ± 0.03	0.56 cde ± 0.02	0.80 c–g ± 0.04	0.86 b–e ± 0.03	0.82 c–f ± 0.04	94.78 ab ± 7.72	95.74 ab ± 6.95	94.88 ab ± 7.42	1.81 bcd ± 0.06	1.69 c–f ± 0.05	1.79 b–e ± 0.08
**Micro-aggregates**															
CK	1.00 j ± 0.00	-	-	0.65 abc ± 0.03	-	-	1.00 ab ± 0.00	-	-	100 e ± 0.00	-	-	1.52 ij ± 0.03	-	-
0.54	1.09 hi ± 0.02	1.07 ij ± 0.04	1.05 ij ± 0.05	0.62 b–f ± 0.04	0.64 bcd ± 0.03	0.65 abc ± 0.02	0.95 b–e ± 0.02	0.98 bc ± 0.03	0.99 ab ± 0.04	105.07 cd ± 5	105.56 cd ± 4	104.57 cde ± 6	1.59 f–i ± 0.05	1.55 g–j ± 0.04	1.53 hij ± 0.09
1.08	1.19 d–g ± 0.03	1.23 c–f ± 0.06	1.25 bcd ± 0.03	0.57 d–h ± 0.02	0.54 ghi ± 0.04	0.53 ghi ± 0.03	0.88 c–g ± 0.04	0.83 fgh ± 0.02	0.80 gh ± 0.03	104.90 cde ± 7	101.73 de ± 6	101.84 de ± 5	1.73 c–g ± 0.05	1.84 b–e ± 0.08	1.87 a–d ± 0.07
1.62	1.29 bc ± 0.06	1.18 d–g ± 0.05	1.26 bcd ± 0.05	0.52 hi ± 0.03	0.58 c–h ± 0.03	0.55 f–i ± 0.02	0.80 gh ± 0.05	0.89 b–g ± 0.04	0.81 fgh ± 0.02	102.36 de ± 8	105.51 cd ± 5	103.24 de ± 8	1.90 abc ± 0.06	1.70 d–h ± 0.09	1.85 a–d ± 0.09
2.16	1.33 ab ± 0.04	1.24 cde ± 0.03	1.38 a ± 0.07	0.51 hi ± 0.04	0.55 f–i ± 0.02	0.49 i ± 0.03	0.78 gh ± 0.04	0.82 fgh ± 0.03	0.75 h ± 0.05	104.74 cde ± 7	103.05 de ± 9	104.23 de ± 8	1.93 ab ± 0.07	1.84 a–e ± 0.06	2.02 a ± 0.08
2.70	1.17 e–h ± 0.05	1.08 hi ± 0.06	1.13 ghi ± 0.06	0.61 b–f ± 0.02	0.71 a ± 0.03	0.62 b–e ± 0.01	0.94 b–e ± 0.04	1.09 a ± 0.02	0.96 bcd ± 0.03	112.02 b ± 8	118.98 a ± 9	109.27 bc ± 8	1.60 f–i ± 0.07	1.40 j ± 0.08	1.57 f–j ± 0.09
GM	1.22 c–f ± 0.04	1.16 fgh ± 0.05	1.21 c–f ± 0.05	0.56 e–h ± 0.03	0.59 b–g ± 0.03	0.55 f–i ± 0.02	0.86 d–g ± 0.04	0.91 b–f ± 0.03	0.84 e–h ± 0.3	105.50 cd ± 7	105.40 cd ± 6	104.10 de ± 7	1.75 b–f ± 0.06	1.66 e–i ± 0.07	1.78 b–e ± 0.08

Averages (mean ± SE) followed by different lowercase letters in the same column are significantly different (Duncan’s, *p* < 0.05) in the same aggregate fraction, *n* = 4. CK: control with no amendment addition; SM: sheep manure; CD: cow dung; SMCD: mixture of sheep manure and cow dung; GM, mean of gradients.

**Table 3 plants-14-01442-t003:** Soil macro-nutrients under different organic manure treatments on alpine meadows ecosystem.

Fertilization Gradients (kg m^−2^)					Treatments				
Soil Aggregate Fraction	SM	CD	SMCD	SM	CD	SMCD	SM	CD	SMCD
Macro-aggregates	TN (g kg ^−1^)			AN (mg kg ^−1^)			TP (g kg ^−1^)		
CK	7.40 k ± 0.20	-	-	514 j ± 10	-	-	0.71 h ± 0.02	-	-
0.54	8.14 ghi ± 0.25	7.97 j ± 0.29	8.17 gh ± 0.24	538 i ± 16	563 h ± 10	595 g ± 15	0.82 g ± 0.03	0.85 fg ± 0.04	0.89 d–g ± 0.02
1.08	8.18 g ± 0.30	8.05 hij ± 0.30	7.97 j ± 0.20	564 h ± 13	590 g ± 12	610 fg ± 18	0.91 c–g ± 0.05	0.90 d–g ± 0.03	0.86 efg ± 0.01
1.62	9.12 ab ± 0.28	8.34 f ± 0.50	9.22 a ± 0.30	602 fg ± 11	635 cde ± 17	655 c ± 12	0.99 abc ± 0.04	1.02 ab ± 0.05	0.97 a–d ± 0.03
2.16	8.99 c ± 0.36	8.07 g–j ± 0.45	9.00 bc ± 0.43	643 cd ± 12	676 b ± 14	704 a ± 20	0.96 a–d ± 0.08	1.00 abc ± 0.07	1.01 a ± 0.05
2.70	8.81 d ± 0.40	8.02 ij ± 0.35	8.54 e ± 0.54	618 ef ± 19	632 de ± 11	650 cd ± 14	0.95 a–e ± 0.06	0.84 fg ± 0.04	0.83 g ± 0.07
GM	8.64 e ± 0.31	8.09 ghi ± 0.36	8.58 e ± 0.32	593 g ± 14	619 ef ± 12	643 cd ± 16	0.93 b–f ± 0.05	0.92 c–g ± 0.04	0.91 c–g ± 0.03
**Micro-aggregates**									
CK	7.34 l ± 0.25	-	-	507 i ± 12	-	-	0.70 h ± 0.03	-	-
0.54	8.10 fh ± 0.50	7.87 hij ± 0.60	7.69 k ± 0.45	532 h ± 15	582 h ± 13	583 g ± 10	0.76 e–h ± 0.04	0.74 gh ± 0.03	0.83 cd ± 0.05
1.08	8.29 e ± 0.40	7.98 ghi ± 0.20	7.85 ij ± 0.30	550 h ± 18	585 g ± 16	601 fg ± 14	0.81 def ± 0.02	0.86 cd ± 0.04	0.73 e–h ± 0.04
1.62	9.01 b ± 0.36	8.15 f ± 0.35	9.15 a ± 0.32	594 fg ± 13	628 cd ± 18	639 c ± 13	1.01 a ± 0.05	0.95 ab ± 0.04	0.84 cd ± 0.01
2.16	8.76 c ± 0.41	7.99 gh ± 0.56	8.88 c ± 0.11	631 c ± 17	661 b ± 20	695 a ± 16	0.92 bc ± 0.07	0.85 cd ± 0.06	1.02 a ± 0.06
2.70	8.59 d ± 0.40	7.80 jk ± 0.25	8.40 e ± 0.36	606 ef ± 20	624 cde ± 15	638 c ± 15	0.81 d–g ± 0.08	0.75 e–h ± 0.08	0.71 fgh ± 0.06
GM	8.55 d ± 0.30	7.95 hi ± 0.32	8.39 e ± 0.47	582 g ± 16	610 def ± 17	631 c ± 13	0.86 cd ± 0.05	0.83 de ± 0.05	0.82 cd ± 0.04
**Macro-aggregates**	**AP (mg kg** ^−^ ^1^ **)**			**TK (g kg** ^−^ ^1^ **)**			**AK (mg kg** ^−^ ^1^ **)**		
CK	17.26 f ± 0.5	-	-	6.76 i ± 0.12	-	-	557 i ± 12	-	-
0.54	19.81 c–f ± 0.6	18.80 ef ± 0.2	22.53 a–d ± 0.7	7.07 h ± 0.17	7.14 gh ± 0.17	7.18 g ± 0.20	583 fgh ± 14	580 gh ± 19	596 a–e ± 18
1.08	20.00 c–f ± 0.7	19.86 c–f ± 0.3	19.43 def ± 0.9	7.15 gh ± 0.22	7.11 gh ± 0.23	7.21 g ± 0.13	584 e–h ± 17	587 d–h ± 14	599 a–d ± 25
1.62	20.73 cde ± 0.9	20.80 b–e ± 0.5	20.70 cde ± 0.5	7.55 de ± 0.08	7.44 ef ± 0.21	7.75 c ± 0.11	606 ab ± 19	609 a ± 12	602 abc ± 11
2.16	24.13 ab ± 0.8	23.00 abc ± 0.6	24.66 a ± 0.6	7.87 b ± 0.15	7.74 c ± 0.16	8.03 a ± 0.18	600 a–d ± 20	587 d–h ± 11	592 c–g ± 16
2.70	23.13 abc ± 0.9	22.06 a–e ± 0.8	19.16 ef ± 0.8	7.88 b ± 0.19	7.65 cd ± 0.18	7.87 b ± 0.14	597 a–e ± 22	579 h ± 23	598 a–e ± 18
GM	21.56 a–e ± 0.8	20.90 b–e ± 0.5	21.30 b–e ± 0.7	7.50 ef ± 0.16	7.41 f ± 0.19	7.60 d ± 0.15	594 b–f ± 18	588 d–h ± 15	597 a–e ± 17
**Micro-aggregates**									
CK	17.00 f ± 0.4	-	-	6.71 h ± 0.10	-	-	550 h ± 8	-	-
0.54	19.30 def ± 0.8	18.36 ef ± 0.7	21.91 abc ± 0.6	6.81 gh ± 0.14	6.87 fg ± 0.20	6.84 fg ± 0.08	576 fg ± 10	567 g ± 13	587 c–f ± 11
1.08	19.33 def ± 0.5	18.90 def ± 0.6	18.26 ef ± 0.7	6.92 ef ± 0.13	6.93 ef ± 0.14	6.94 ef ± 0.17	586 c–f ± 12	581 efg ± 16	596 abcd ± 17
1.62	20.06 b–e ± 0.7	19.49 cde ± 0.9	18.70 def ± 0.9	6.98 e ± 0.19	7.09 d ± 0.11	7.11 d ± 0.19	600 abc ± 17	590 b–f ± 19	607 a ± 14
2.16	23.76 a ± 0.6	21.10 bcd ± 0.7	20.26 b–e ± 0.9	7.30 c ± 0.20	7.42 b ± 0.09	7.28 c ± 0.22	588 c–f ± 20	583 def ± 15	585 c–f ± 18
2.70	22.02 ab ± 0.8	21.00 bcd ± 0.4	18.13 ef ± 0.5	7.50 b ± 0.16	7.65 a ± 0.16	7.75 a ± 0.12	604 ab ± 16	578 efg ± 10	580 efg ± 14
GM	20.89 bcd ± 0.7	19.77 b–e ± 0.6	19.45 c–f ± 0.6	7.10 d ± 0.16	7.19 d ± 0.14	7.17 d ± 0.15	591 b–e ± 15	580 efg ± 14	590b–e ± 14

Different lowercase letters in the similar aggregate fraction show significant differences between different treatments at *p* < 0.05 (Duncan’s test was performed for mean separation). Values are means ± SE (*n* = 4). The abbreviated words stand for TN, total nitrogen; AN, available nitrogen; TP, total phosphorous; AP, available phosphorous; TK, total potassium; AK, available potassium. CK: unamended control; SM: sheep manure; CD: cow dung; SMCD: mixture of sheep manure and cow dung: GM, gradient mean.

**Table 4 plants-14-01442-t004:** Pre-fertilization basic properties of 0–10 cm soil depth at the alpine meadow in Tianzhu county of Gansu China.

Soil Parameter	Values	Measurement Method	References
SOC (g kg^−1^)	3.20 ± 78.3	Walkley–Black dichromate oxidation	Nelson and Sommers [92]
TN (g kg^−1^)	0.50 ± 7.12	Semimicro-Kjeldahl method	Lu [93]
TP (g kg^−1^)	0.04 ± 0.70	Colorimetric method	Lu [93]
TK (g kg^−1^)	0.33 ± 6.55	Colorimetric method	Lu [93]
pH	0.22 ± 7.8	pH meter	Lu [93]
BD (g cm^−3^)	0.03 ± 0.78	Core sampler method	Bao [94]
P (%)	0.40 ± 70.8	(1 − (BD/PD)) × 100 equation	Bao [94]

The abbreviated words stand for SOC, soil organic carbon; TN, total nitrogen; TP, total phosphorous; TK, total potassium; pH, soil pH; BD, soil bulk density; P, soil pore space; PD, soil particle density = (2.65 g cm^−3^).

**Table 5 plants-14-01442-t005:** Parameter description used for analyzing micronutrients with Atomic Absorption Spectrometer (SPECTRAA2207).

Parameters	Zn	Cu	Fe	Mn
Current of lamp (mA)	10	7.5	10	3
Wavelength analysis (nm)	213.9	324.8	248.3	279.5
Width of slit (nm)	1.3	1.3	0.2	0.5
Type of atomizer flame	Standard burner air-acetylene	Standard burner air-acetylene	Standard burner air-acetylene	Standard burner air-acetylene
Velocity of gas (L/h)	1.6	1.6	1.6	50
Type of consumption head (mm)	0.2	0.3	0.3	100
Height of consumption head (mm)	7.5	7.5	7.5	6

## Data Availability

No new data were created or analyzed in this study. Data sharing is not applicable to this article.

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
