# Peer review of "Soil Bulk Density, Aggregates, Carbon Stabilization, Nutrients and Vegetation Traits as Affected by Manure Gradients Regimes Under Alpine Meadows of Qinghai–Tibetan Plateau Ecosystem"

_plants, 2025, doi:10.3390/plants14101442_

Round 1
Reviewer 1 Report
Comments and Suggestions for Authors
Title of the Manuscript: Soil Bulk Density, Aggregates, Carbon Stabilization, Nutrients, and Vegetation Traits as Affected by Manure Gradient Regimes in the Alpine Meadows of the Qinghai-Tibetan Plateau
This manuscript addresses a significant topic related to soil carbon sequestration, nutrient dynamics, and vegetation characteristics under varying organic manure regimes in the Qinghai-Tibetan Plateau. The study’s relevance to ecological sustainability and grassland management is evident, and the three-year field experiment provides valuable insights. However, several aspects require clarification and improvement to enhance the overall quality of the manuscript and ensure the clarity and reliability of the research findings.
Specific Comments:
- Clarity and Structure:
The structure of the manuscript could be clearer, especially in the "Introduction" and "Results" sections. The introduction contains excessive information that could be presented more concisely to maintain reader engagement.
Use subheadings in the results section to better organize the findings (e.g., soil bulk density, SOC fractions, nutrient dynamics, vegetation traits).
- Language and Grammar:
The manuscript contains several grammatical errors and unclear phrasing, which impede readability. For example, the phrase "more exploration" could be replaced with "further investigation." A thorough language review is recommended to ensure clarity and consistency.
- Methodology:
The experimental design, particularly the rationale for selecting specific manure input gradients, should be clearly justified. Why were the gradients of 0.54, 1.08, 1.62, 2.16, and 2.70 kg m² chosen?
Provide more detailed descriptions of the Randomized Complete Block Design (RCBD) implementation, including plot size and replication strategy.
The description of the analytical methods for SOC fractions and nutrient measurements should be more detailed, including references to standard protocols or any modifications made.
- Data Presentation:
The tables and figures provide valuable information but should be more effectively integrated into the text. Clearly refer to each figure/table in the corresponding results subsection.
Ensure consistent use of units and notation throughout the manuscript. For example, "kg m²" should be formatted consistently.
- Results Interpretation:
The discussion on the differential effects of sheep manure (SM), cow dung (CD), and their mixture (SMCD) on SOC fractions and nutrient dynamics is insightful but could be further elaborated. Consider discussing potential mechanisms driving the observed patterns.
The negative response of the Carbon Management Index (CMI) in macro aggregates is noted but not sufficiently explored. Providing a hypothesis or referring to existing literature could strengthen this section.
- References and Citations:
Ensure all claims, particularly those regarding previous studies and general knowledge statements, are appropriately cited.
Cross-check the reference list for completeness and formatting consistency.
- Conclusion:
The conclusion effectively summarizes the findings but could benefit from a clearer statement on the implications for grassland management policies and practices.
Consider suggesting avenues for future research based on the study’s findings and limitations.
Minor Issues:
- Line 15: "Climate change and overgrazing harshly constrain meadows land sustainability..." Consider rephrasing to "Climate change and overgrazing significantly constrain the sustainability of meadowlands..."
- Line 48: "The alpine meadows should be managed with manure in SM under 2.16 kg m⁻²..." Consider rephrasing for clarity.
Recommendation:
The manuscript presents valuable research findings but requires major revisions to improve clarity, methodological transparency, and data interpretation. I recommend major revision before it can be considered for publication.
Comments on the Quality of English Language
Language and Grammar:
The manuscript contains several grammatical errors and unclear phrasing, which impede readability. For example, the phrase "more exploration" could be replaced with "further investigation." A thorough language review is recommended to ensure clarity and consistency.
Author Response
Authors Response for reviewers 1
Cover Letter and Author Response for Manuscript [Soil bulk density, aggregates, carbon stabilization, nutrients and vegetation traits as affected by manures gradients regimes under alpine meadows of Qinghai-Tibetan Plateau ecosystem].
March 2, 2025
The Revised areas are in blue font.
Dear respected editor,
We thank you and the reviewers for your encouraging comments on our manuscript entitled [Soil bulk density, aggregates, carbon stabilization, nutrients and vegetation traits as affected by manures gradients regimes under alpine meadows of Qinghai-Tibetan Plateau ecosystem]. We appreciate the opportunity to revise and improve this manuscript to a level suitable for publication. We carefully considered all reviewer comments and made point by point revisions to the manuscript to address these comments. Our responses to the reviewer comments are listed below in the blue font and are shown in blue font in the revised manuscript.
We hope our efforts have addressed all of the concerns to level required for publication. Thank you very much for your help. We look forward to hearing from you. Please contact us if there are questions or concerns.
Thank you very much.
Sincerely’
Authors:
Mahran Sadiq1,2,3,, Nasir Rahim2, Majid Mahmood Tahir2, Aqila Shaheen2, M Ajmal Ali3, Mohamed S. Elshikh3, Fu Ran1, Guoxiang Chen1 and Bai Xiaoming1,4*
1College of Grassland Science, Gansu Agricultural University, Lanzhou 730070, China; khanmahran420@gmail.com (M.S.)
2Department of Soil and Environmental Sciences, University of Poonch Rawalakot, AJK 12350, Pakistan; majidmahmood@upr.edu.pk (M.M.T.)
3Department of Botany and Microbiology, College of Science, King Saud University, Riyadh 11451, Saudi Arabia
4Key Laboratory of Grassland Ecosystem (Ministry of Education), Lanzhou 730070, China
*Correspondence: baixm@gsau.edu.cn
Responses to Reviewers Comments
This manuscript addresses a significant topic related to soil carbon sequestration, nutrient dynamics, and vegetation characteristics under varying organic manure regimes in the Qinghai-Tibetan Plateau. The study’s relevance to ecological sustainability and grassland management is evident, and the three-year field experiment provides valuable insights. However, several aspects require clarification and improvement to enhance the overall quality of the manuscript and ensure the clarity and reliability of the research findings.
Response: Bundle thanks for encouraging.
Specific Comments:
- Clarity and Structure:
The structure of the manuscript could be clearer, especially in the "Introduction" and "Results" sections. The introduction contains excessive information that could be presented more concisely to maintain reader engagement.
Response: The introduction contained detail information related with problems and significance of climate change, carbon sequestration, soil properties, meadows vegetation traits and manures. Actually, number of parameters are studied in this manuscript and even little information about each section makes the lengthy intro. Now its modified according to respected reviewers suggestions, Now, introduction section is summarized with short and relevant information’s on climate change, carbon sequestration, soil properties, meadows vegetation traits and manures in order to maintain reader engagement and additions are in blue font in the manuscript.
Use subheadings in the results section to better organize the findings (e.g., soil bulk density, SOC fractions, nutrient dynamics, vegetation traits).
Response: We respect the suggestion of respected reviewer, however, subheadings in different sections soil bulk density and aggregate size distribution, nutrient dynamics, Forage biodiversity, productivity and nutritional quality are more specific than soil bulk density, SOC fractions, nutrient dynamics, vegetation traits, because these are general terms. Specific terminologies are better for readability and understanding for future scholars.
- Language and Grammar:
The manuscript contains several grammatical errors and unclear phrasing, which impede readability. For example, the phrase "more exploration" could be replaced with "further investigation." A thorough language review is recommended to ensure clarity and consistency.
Response: Now manuscript has revised thoroughly, each and word we used in this article is very very clear and professional in the field of soil and environmental sciences, which increase the readability for coming scientists. Moreover, more exploration is now replaced by further investigation according to reviewers suggestions. All modifications are in blue font at manuscript.
- Methodology:
The experimental design, particularly the rationale for selecting specific manure input gradients, should be clearly justified. Why were the gradients of 0.54, 1.08, 1.62, 2.16, and 2.70 kg m² chosen?
Provide more detailed descriptions of the Randomized Complete Block Design (RCBD) implementation, including plot size and replication strategy.
Response: The present field study of manures application under different gradients practices was conducted using a randomized complete block design (RCBD) with four replications during March to August of 2022-2024. The research comprised of four treatments counting one un-amended (CK) control and three diverse manures practices specifically., alone sheep manure (SM), sole cow dung (CD) and mixture of 50% sheep manure and 50% cow dung (SMCD) with five gradients i.e., 0.54, 1.08, 1.62, 2.16 and 2.70 kg m-2 using 7.50, 15, 22.50, 30, 37.50 g m-2 nitrogen rates. All experimental treatments were implemented in 64 plots including four replicates of each treatment. Each individually plot covered an area of 30 m2 (6 m × 5 m), and a 1-m buffer was set between two adjacent plots without manure application. All information regarding RCBD) implementation, including plot size and replication strategy and manure input gradients are added in the manuscript.
The description of the analytical methods for SOC fractions and nutrient measurements should be more detailed, including references to standard protocols or any modifications made.
Response: The description of the analytical methods for SOC fractions and nutrient measurements are now added according to reviewers suggestions. All modifications are in blue font at manuscript.
- Data Presentation:
The tables and figures provide valuable information but should be more effectively integrated into the text. Clearly refer to each figure/table in the corresponding results subsection.
Response: Now all tables and figures are very clearly referred in the text according to reviewers suggestions. All modifications are in blue font at manuscript.
Ensure consistent use of units and notation throughout the manuscript. For example, "kg m²" should be formatted consistently.
Response: All units are consistent throughout the manuscript.
- Results Interpretation:
The discussion on the differential effects of sheep manure (SM), cow dung (CD), and their mixture (SMCD) on SOC fractions and nutrient dynamics is insightful but could be further elaborated. Consider discussing potential mechanisms driving the observed patterns.
Response: The possible reasons or mechanisms driving the observed patterns are clearly mention in different sections of discussion.
The negative response of the Carbon Management Index (CMI) in macro aggregates is noted but not sufficiently explored. Providing a hypothesis or referring to existing literature could strengthen this section.
Response: Now this section has revised and additions are in blue font in the text. All driving factors and mechanisms are highlighted in the text regarding different parameters. Our research demonstrated that, the effects of manures input rates on CMI and SI were noteworthy with CMI decreasing and SI increasing in treatments receiving manures input rates than CK at macro aggregates. Here, we plan a mechanism in which manures addition with contrasting inorganic nitrogen management practice may help to regulate SOC dynamics in Cfrac1, Cfrac2 and Cfrac3 and also protect current SOC from further mineralization. Consequently, both manures and inorganic nitrogen will increase the soil CMI at macro aggregates. However, both CMI and SI were increased under manures addition in case of micro aggregates. This is confirmed by several scholars in their long-term fertilization studies. The CMI was extremely correlated with the amount of total SOC and each labile and non-labile C fraction. Tirol-Padre and Ladha, clarified that CMI differences under fertilizer treatments are credited to the rise in C addition and the organic matter quality variations, thus influencing the C susceptibility to oxidation of KMnO4. Under manures input rates, the rise of SI was owing to the rate of labile C loss being higher than that of stable C. This is mostly due to the differential microorganism’s accessibility to diverse active C fractions. The complex structure of stable carbon sheltered it against bacterial attack. Moreover, we found that the SI under manures addition of macro aggregates was higher than that of micro aggregates, might be due to greater SOC protection ability. Carbon storage changes related with macro aggregate fraction were suggestively affected by manures addition over micro aggregate fraction. This specified that the dynamic changes in the C pool are more sensitive in the macro aggregates compared with micro aggregates. Therefore, manures input rates improved the soil C pools stability by stimulating bacterial active nutrients utilization and altering forage litter quality, and macro aggregates C pools are more sensitive to manures addition.
- References and Citations:
Ensure all claims, particularly those regarding previous studies and general knowledge statements, are appropriately cited.
Cross-check the reference list for completeness and formatting consistency.
Response: All references are appropriately cited in the text are mention in the references section. All references are complete with MDPI formate.
- Conclusion:
The conclusion effectively summarizes the findings but could benefit from a clearer statement on the implications for grassland management policies and practices.
Consider suggesting avenues for future research based on the study’s findings and limitations.
Response: Our research concludes that the application of sole SM under 2.16 kg m-2 input rate regime leads to the enhancement of the carbon stabilization, nutrients accumulation, meadows productivity and forage quality. Therefore, this research highlights the need for sustainable meadows management that can facilitate the more sequestration of carbon, which eventually increases the soil quality, vegetation traits and ecological balance, and ultimately serve the global sustainability purpose. The addition of organic manures is more efficient in increasing carbon stabilization, physico-chemical properties and vegetation traits. In essence, our research significantly contributes to understanding the effect of organic manures under varying input rates on soil organic carbon stability, soil and forage quality as well as productivity, offering valuable insights for mitigating climate change and overgrazing and managing soil, environmental and vegetation quality in alpine meadow ecosystem. As such for sustaining alpine meadow sustainability in Tianzhu, China, SM under 2.16 kg m-2 input rate is a forward-thinking and ecologically sound practice, should be recommended and promoted amongst the household herdsmen’s. This research will also be supportive for the grassland use policy makers to strategically plan suitable management/restoration techniques for ecological grassland use management in extremely frangible alpine meadow ecosystems and other analogous places in the world. All modification are in blue font in the text.
Minor Issues:
- Line 15: "Climate change and overgrazing harshly constrain meadows land sustainability..." Consider rephrasing to "Climate change and overgrazing significantly constrain the sustainability of meadowlands..."
Response: Now its revised according to respected reviewers suggestions as rephrased Climate change and overgrazing significantly constrain the sustainability of meadowlands. All modifications are in blue font at manuscript.
- Line 48: "The alpine meadows should be managed with manure in SM under 2.16 kg m⁻²..." Consider rephrasing for clarity.
Response: Now its revised according to respected reviewers suggestions. All modifications are in blue font at manuscript.
Recommendation:
The manuscript presents valuable research findings but requires major revisions to improve clarity, methodological transparency, and data interpretation. I recommend major revision before it can be considered for publication.
Response: We carefully considered all reviewer comments and made point by point revisions to the manuscript to address these comments. We hope our efforts have addressed all of the concerns to level required for publication. Thank you very much for your help.
Language and Grammar:
The manuscript contains several grammatical errors and unclear phrasing, which impede readability. For example, the phrase "more exploration" could be replaced with "further investigation." A thorough language review is recommended to ensure clarity and consistency.
Response: We thoroughly revised the article and used most relevant and clear word in the field of soil and environmental sciences for better readability. We hope our efforts have addressed all of the concerns to level required for publication.
Thank you very much.
Sincerely’
Authors

Reviewer 2 Report
Comments and Suggestions for Authors
Comments and Suggestions for Authors
Title: Soil bulk density, aggregates, carbon stabilization, nutrients and
vegetation traits as affected by manures gradients regimes under alpine
meadows of Qinghai-Tibetan Plateau ecosystem
Dear Authors
The research results presented in the manuscript fall within the publishing profile of Plants.
The authors determined the effect of organic fertilizers (sheep manure, cow manure, a mixture of sheep manure and cow manure) on the size of soil aggregates, the pools of organic carbon associated with soil aggregates and their stabilization, nutrients, vegetation structure, biomass and nutritional quality of feed.
The manuscript, although voluminous, is generally well written. The authors well presented the research objectives and research hypotheses. The obtained research results are local in scope, but due to the common soil type they may have a global character.
Multidirectional and extensive research and the obtained results constitute a source of knowledge regarding the possibilities of using manures on alpine meadows for the purpose of ecological use of the meadow ecosystem.
The authors proposed specific doses of manures for application to alpine meadows. The research results were described in detail and properly discussed in the Discussion section. Conclusions formulated correctly.
The references used in the manuscript can be considered appropriate.
In order to increase the usefulness of the article, Authors must refer to the following points.
Remarks
- Abstract - Line 27 - Please provide manure rates in t ha-1 or Mg ha-1. This note applies to the entire manuscript. When using abbreviations for the first time, provide full names, e.g. BD (L.29).
- Keywords - Consider removing the word "gradients".
- The Introduction section is too long. Please consider shortening this section. Line 144 Mg, Ca, S - These are not micronutrients.
- Results: The research results were well presented and described but require improvement. In the entire section, the unit for organic fertilizer (manure) doses should be corrected and of course recalculated quantitatively. It should be: t ha-1 or Mg ha-1. This note applies to the entire manuscript. After all, this was field research. The readability of the statistical description in Figures 1, 2, 3, and 4 should be improved. Letter symbols in Tables 1-3 should be placed next to the numerical data relating to the parameters being marked and not next to the SE values. Lines 524-525 The abbreviations given in Figures 4e-h should be added to the explanations.
- Discussion: L.595-597 Should be removed. No such studies have been conducted.
- Materials and Methods: Subsection 4.1. The distribution of precipitation and temperatures during the growing seasons of the studies conducted should be added. Subsection 4.5. The patterns used should be numbered, e.g. (1), (2), etc. Line 912 and 940 - The order of citations or the References list should be corrected. For listed equipment please provide the name, manufacturing company, country and city. Why were plant samples taken only in 2024 (L.947) and soil samples taken in 2022-2024 (L.876)?
- Conclusion: Do the Authors see a need to continue research? If so, please add a direction for further research.
Best regards
Author Response
Authors Response for reviewers 2
Cover Letter and Author Response for Manuscript [Soil bulk density, aggregates, carbon stabilization, nutrients and vegetation traits as affected by manures gradients regimes under alpine meadows of Qinghai-Tibetan Plateau ecosystem].
March 2, 2025
The Revised areas are in blue font.
Dear respected editor,
We thank you and the reviewers for your encouraging comments on our manuscript entitled [Soil bulk density, aggregates, carbon stabilization, nutrients and vegetation traits as affected by manures gradients regimes under alpine meadows of Qinghai-Tibetan Plateau ecosystem]. We appreciate the opportunity to revise and improve this manuscript to a level suitable for publication. We carefully considered all reviewer comments and made point by point revisions to the manuscript to address these comments. Our responses to the reviewer comments are listed below in the blue font and are shown in blue font in the revised manuscript.
We hope our efforts have addressed all of the concerns to level required for publication. Thank you very much for your help. We look forward to hearing from you. Please contact us if there are questions or concerns.
Thank you very much.
Sincerely’
Authors:
Mahran Sadiq1,2,3,, Nasir Rahim2, Majid Mahmood Tahir2, Aqila Shaheen2, M Ajmal Ali3, Mohamed S. Elshikh3, Fu Ran1, Guoxiang Chen1 and Bai Xiaoming1,4*
1College of Grassland Science, Gansu Agricultural University, Lanzhou 730070, China; khanmahran420@gmail.com (M.S.)
2Department of Soil and Environmental Sciences, University of Poonch Rawalakot, AJK 12350, Pakistan; majidmahmood@upr.edu.pk (M.M.T.)
3Department of Botany and Microbiology, College of Science, King Saud University, Riyadh 11451, Saudi Arabia
4Key Laboratory of Grassland Ecosystem (Ministry of Education), Lanzhou 730070, China
*Correspondence: baixm@gsau.edu.cn
Responses to Reviewers Comments
Dear Authors
The research results presented in the manuscript fall within the publishing profile of Plants.
The authors determined the effect of organic fertilizers (sheep manure, cow manure, a mixture of sheep manure and cow manure) on the size of soil aggregates, the pools of organic carbon associated with soil aggregates and their stabilization, nutrients, vegetation structure, biomass and nutritional quality of feed.
The manuscript, although voluminous, is generally well written. The authors well presented the research objectives and research hypotheses. The obtained research results are local in scope, but due to the common soil type they may have a global character.
Multidirectional and extensive research and the obtained results constitute a source of knowledge regarding the possibilities of using manures on alpine meadows for the purpose of ecological use of the meadow ecosystem.
The authors proposed specific doses of manures for application to alpine meadows. The research results were described in detail and properly discussed in the Discussion section. Conclusions formulated correctly.
The references used in the manuscript can be considered appropriate.
In order to increase the usefulness of the article, Authors must refer to the following points.
Response: Bundle thanks for encouraging.
The need of futher research has been explianed under limition section of paper.
Remarks
- Abstract - Line 27 - Please provide manure rates in t ha-1 or Mg ha-1. This note applies to the entire manuscript. When using abbreviations for the first time, provide full names, e.g. BD (L.29).
Response: We respect the suggestion of respected reviewer, however, in order to keep the uniformity of manure input rates with our other published research paper, we would like to keep same unit, Many thanks for your understanding. Now modifications in abstract has done according to respected reviewers suggestions as full name of BD has added. All modifications are in blue font at manuscript.
- Keywords - Consider removing the word "gradients".
Response: Now keywords are revised according to respected reviewers suggestions as gradients has deleted.
- The Introduction section is too long. Please consider shortening this section. Line 144 Mg, Ca, S - These are not micronutrients.
Response: The introduction contained information in detail related with problems and significance of climate change, carbon sequestration, soil properties, meadows vegetation traits and manures. Actually, number of parameters are studied in this study and even little information about each parameter makes the lengthy intro. Now its modified according to respected reviewers suggestions, Now, introduction section is summarized with short information’s on climate change, carbon sequestration, soil properties, meadows vegetation traits and manures. Also, according to your suggestion Mg, Ca, S are removed from article.
- Results: The research results were well presented and described but require improvement. In the entire section, the unit for organic fertilizer (manure) doses should be corrected and of course recalculated quantitatively. It should be: t ha-1 or Mg ha-1. This note applies to the entire manuscript. After all, this was field research. The readability of the statistical description in Figures 1, 2, 3, and 4 should be improved. Letter symbols in Tables 1-3 should be placed next to the numerical data relating to the parameters being marked and not next to the SE values. Lines 524-525 The abbreviations given in Figures 4e-h should be added to the explanations.
Response: The result section contains 31 parameters and the most important information’s are mentioned in this section. All soil indicators are tested at soil aggregate level and although this section is also voluminous but info regarding both aggregates separately is important for readers. All figures have 300dpi size with good quality. The number of bars in the figures are many thatswhy width of bars are small because our treatments are many, each figure contains 19 bars thatswhy all bars are closely connected. However, all figures are in good quality with 300 dpi size. Moreover, all letter symbols are now revised according to respected reviewers suggestion. All figures are explained in the text, additions are in manuscript.
- Discussion: L.595-597 Should be removed. No such studies have been conducted.
Response: Now its revised in accordance with reviewers suggestion. All modifications are in blue font at manuscript.
- Materials and Methods: Subsection 4.1. The distribution of precipitation and temperatures during the growing seasons of the studies conducted should be added. Subsection 4.5. The patterns used should be numbered, e.g. (1), (2), etc. Line 912 and 940 - The order of citations or the References list should be corrected. For listed equipment please provide the name, manufacturing company, country and city. Why were plant samples taken only in 2024 (L.947) and soil samples taken in 2022-2024 (L.876)?
Response: The climatic data was not recorded during this study time, but in the future we will record the climatic data on daily basis. Many thanks for you comments, it will make our scientific projects better. Subsection 4.5. Now this sections has revised and in accordance with reviewers suggestion. All modifications are in blue font at manuscript. All references are now correct and listed in the article text. Now the information of equipment has added in accordance with reviewers suggestion. The manures or organic amendments causes the significant changes after application of more than 2 years, as noted by many scholars globally thatswhy we had plan to noted the vegetation parameters after 3 yrs application of manures. Moreover, our study mainly focused on soil carbon contents thatswhy we noted changes in carbon and nutrients and soil physical indictors for 3 yrs.
- Conclusion: Do the Authors see a need to continue research? If so, please add a direction for further research.
This research highlights the need for sustainable meadows management that can facilitate the more sequestration of carbon, which eventually increases the soil quality, vegetation traits and ecological balance, and ultimately serve the global sustainability purpose. All modifications are incorporated in the manuscript.
Thank you very much.
Sincerely’
Authors

Reviewer 3 Report
Comments and Suggestions for Authors
Comments to the paper “Soil bulk density, aggregates, carbon stabilization, nutrients and vegetation traits as affected by manures gradients regimes under alpine meadows of Qinghai-Tibetan Plateau ecosystem”
General Comment
The submitted document is well structured and provides a sufficiently in-depth analysis of the impact of manure on high altitude meadows in the Himalayan mountains. However, some sections should be made more concise and more focused to improve the readability and flow of the text. The experimental design is good and oriented towards the practicality of the interventions although limited to only three years of experimentation that are not definitive in the evaluation of the long-term effects of animal excretions on meadows. This study is good but very specific for the Tibetian grasslands and may be not simply applicable in other alpine areas on the planet.
Abstract
The abstract gives a concise and clear description of the objectives as well as the methods used, and the results obtained. It reports how important it is to properly manage the alpine meadows at these high altitudes with the use of manure. However, it would be appropriate and convenient to explain in the text of this section the practical usefulness of the results obtained with the three-year experiment. # ). Abbreviations in the abstract should be made explicit e.g. peds = macro-aggregate fraction).
Introduction
The section is too long and cumbersome with some unnecessary references (see e.g. references about odor problems) that would require more conciseness and only the most pertinent references. However, the study is well described with references to climate change and the problems arising from overgrazing. The reasons and purposes of the research are well highlighted.
Material and methods
The experimental design is correct and relevant. Sampling as well as analysis methodologies are well structured. The study can be well replicated in high altitude environments of the zones.
Result and discussion
This section is quite dense and the authors could provide a clearer and simpler summary of the most important results obtained to facilitate the readers‘ understanding. However, the results are well explained with good and effective tables and figures. The statistical elaborations are also well done.
Conclusions
The work highlights very well the most important results and also practical aspects derived from them. In the discussion practical information is also provided on how alpine meadows should be managed by organic fertilization. In the following section it would also be advisable to provide more information and discuss how to use the information obtained from the work in other fields in the future. The practical indication of the use of sheep manure for improving the quality of soil and grass is appreciable.
Author Response
Authors Response for reviewers 3
Cover Letter and Author Response for Manuscript [Soil bulk density, aggregates, carbon stabilization, nutrients and vegetation traits as affected by manures gradients regimes under alpine meadows of Qinghai-Tibetan Plateau ecosystem].
March 2, 2025
The Revised areas are in blue font.
Dear respected editor,
We thank you and the reviewers for your encouraging comments on our manuscript entitled [Soil bulk density, aggregates, carbon stabilization, nutrients and vegetation traits as affected by manures gradients regimes under alpine meadows of Qinghai-Tibetan Plateau ecosystem]. We appreciate the opportunity to revise and improve this manuscript to a level suitable for publication. We carefully considered all reviewer comments and made point by point revisions to the manuscript to address these comments. Our responses to the reviewer comments are listed below in the blue font and are shown in blue font in the revised manuscript.
We hope our efforts have addressed all of the concerns to level required for publication. Thank you very much for your help. We look forward to hearing from you. Please contact us if there are questions or concerns.
Thank you very much.
Sincerely’
Authors:
Mahran Sadiq1,2,3,, Nasir Rahim2, Majid Mahmood Tahir2, Aqila Shaheen2, M Ajmal Ali3, Mohamed S. Elshikh3, Fu Ran1, Guoxiang Chen1 and Bai Xiaoming1,4*
1College of Grassland Science, Gansu Agricultural University, Lanzhou 730070, China; khanmahran420@gmail.com (M.S.)
2Department of Soil and Environmental Sciences, University of Poonch Rawalakot, AJK 12350, Pakistan; majidmahmood@upr.edu.pk (M.M.T.)
3Department of Botany and Microbiology, College of Science, King Saud University, Riyadh 11451, Saudi Arabia
4Key Laboratory of Grassland Ecosystem (Ministry of Education), Lanzhou 730070, China
*Correspondence: baixm@gsau.edu.cn
Responses to Reviewers Comments
General Comment
The submitted document is well structured and provides a sufficiently in-depth analysis of the impact of manure on high altitude meadows in the Himalayan mountains. However, some sections should be made more concise and more focused to improve the readability and flow of the text. The experimental design is good and oriented towards the practicality of the interventions although limited to only three years of experimentation that are not definitive in the evaluation of the long-term effects of animal excretions on meadows. This study is good but very specific for the Tibetan grasslands and may be not simply applicable in other alpine areas on the planet.
Response: Bundle thanks for encouraging.
Abstract
The abstract gives a concise and clear description of the objectives as well as the methods used, and the results obtained. It reports how important it is to properly manage the alpine meadows at these high altitudes with the use of manure. However, it would be appropriate and convenient to explain in the text of this section the practical usefulness of the results obtained with the three-year experiment. # ). Abbreviations in the abstract should be made explicit e.g. peds = macro-aggregate fraction).
Response: Now modifications in abstract has done according to respected reviewers suggestions. All modifications are in blue font at manuscript. Overall, manures addition under varying input rates improved the plant community structure, enhanced meadows yield, plant community diversity and nutritional quality than CK. Therefore, the alpine meadows should be managed sustainably by the adoption of sole SM practice under 2.16 kg m-2 input rate for the ecological utilization of meadows ecosystem. The results of this study deliver an innovative perspective in understanding the response of alpine meadows SOC-pools, SOC-stabilization and nutrients at aggregate level, vegetation structure, productivity and forage nutritional quality to manures input rates practices. Moreover, this study provides valuable information for ensuring the climate change mitigation and clean production of alpine meadows in a Qinghai-Tibetan Plateau area in China. Also, abbreviation macroaggregates over microaggregates has modified.
Introduction
The section is too long and cumbersome with some unnecessary references (see e.g. references about odor problems) that would require more conciseness and only the most pertinent references. However, the study is well described with references to climate change and the problems arising from overgrazing. The reasons and purposes of the research are well highlighted.
Response: The introduction contained information in detail related with problems and significance of climate change, carbon sequestration, soil properties, meadows vegetation traits and manures. Actually, number of parameters are studied in this manuscript and even little information about each section makes the lengthy intro. Now its modified according to respected reviewers suggestions, Now, introduction section is summarized with short information’s on climate change, carbon sequestration, soil properties, meadows vegetation traits and manures. References about odor problems are now short with general statement of manures side-effects in agricultural or grassland ecosystems.
Material and methods
The experimental design is correct and relevant. Sampling as well as analysis methodologies are well structured. The study can be well replicated in high altitude environments of the zones.
Response: Many thanks.
Result and discussion
This section is quite dense and the authors could provide a clearer and simpler summary of the most important results obtained to facilitate the readers‘ understanding. However, the results are well explained with good and effective tables and figures. The statistical elaborations are also well done.
Response: Our article contains total of 31 parameters and these parameter are divided into different sections. The results contained the most important info, as highest and lowest values of different amendments used in the experiment at aggregate level. Its very important to describe parameters regarding aggregates, because other scholars mostly studied at bulk soil and this study especially at aggregate level has not studied yet in Tianzhu of China, although its voluminous, but it contains all information for better readability and understanding for future scholars. Each and every section has clear statement with professional wording in soil and environmental sciences for better readability. However, now, its revised and additions are in blue font in the manuscript.
Conclusions
The work highlights very well the most important results and also practical aspects derived from them. In the discussion practical information is also provided on how alpine meadows should be managed by organic fertilization. In the following section it would also be advisable to provide more information and discuss how to use the information obtained from the work in other fields in the future. The practical indication of the use of sheep manure for improving the quality of soil and grass is appreciable.
Response: Now modifications has done according to respected reviewers suggestions. All modifications are in blue font at manuscript. Our research concludes that the application of sole SM under 2.16 kg m-2 input rate regime leads to the enhancement of the carbon stabilization, nutrients accumulation, meadows productivity and forage quality. Therefore, this research highlights the need for sustainable meadows management that can facilitate the more sequestration of carbon, which eventually increases the soil quality, vegetation traits and ecological balance, and ultimately serve the global sustainability purpose. The addition of organic manures is more efficient in increasing carbon stabilization, physico-chemical properties and vegetation traits. In essence, our research significantly contributes to understanding the effect of organic manures under varying input rates on soil organic carbon stability, soil and forage quality as well as productivity, offering valuable insights for mitigating climate change and overgrazing and managing soil, environmental and vegetation quality in alpine meadow ecosystem. As such for sustaining alpine meadow sustainability in Tianzhu, China, SM under 2.16 kg m-2 input rate is a forward-thinking and ecologically sound practice, should be recommended and promoted amongst the household herdsmen’s. This research will also be supportive for the grassland use policy makers to strategically plan suitable management/restoration techniques for ecological grassland use management in extremely frangible alpine meadow ecosystems and other analogous places in the world.
Thank you very much.
Sincerely’
Authors

Round 2
Reviewer 1 Report
Comments and Suggestions for Authors
After reviewing the revised manuscript "Soil bulk density, aggregates, carbon stabilization, nutrients and vegetation traits as affected by manure gradient regimes under alpine meadows of the Qinghai-Tibetan Plateau ecosystem", it is evident that the authors have made substantial improvements based on the reviewers' comments. The structure of the manuscript is now clearer, the introduction has been streamlined to focus on key research aspects, and the results section has been better organized into logical subsections, making it easier for the reader to follow the experimental results and their interpretation.
The scientific content is well presented and coherent, providing a comprehensive discussion of climate change impacts on meadow ecosystems, soil organic carbon (SOC) stabilization, vegetation structure, and nutrient dynamics under different organic manure gradients. The experimental section has been significantly improved, with clear justification for the selection of specific manure application rates (0.54, 1.08, 1.62, 2.16, and 2.70 kg/m²). In addition, the authors have provided a more detailed explanation of the randomized complete block design (RCBD), plot size, replication strategy, and analytical methods used to measure SOC fractions and nutrient levels.
Data presentation has also been improved, with figures and tables more effectively referenced within the text. However, some figures still contain excessive information, making them somewhat complex to interpret at first glance. The simplification of figure annotations and captions would further improve clarity and accessibility.
The manuscript has undergone significant grammatical and linguistic improvements, making it more readable. Ambiguous phrases have been corrected, and technical terminology is now more precise. However, some sections, particularly in the Results and Discussion, still contain long and complex sentences that could be further refined to improve readability.
A key improvement is the expanded discussion of the negative response of the Carbon Management Index (CMI) in macroaggregates, with a clearer explanation of the underlying mechanisms and supporting references. The conclusion section has also been improved, with a stronger statement on the implications of the study for grassland management, environmental sustainability and climate change mitigation. However, highlighting specific recommendations for practical applications and future research directions would further strengthen this section.
The manuscript has been significantly improved and most of the reviewers' concerns have been adequately addressed. However, minor revisions are still needed, particularly to refine complex sentences, simplify data visualization, and further emphasize the practical implications of the study. I recommend publication after minor revisions focusing on language refinement, simplification of figures, and improved clarity of conclusions.
Comments on the Quality of English LanguageThe English in the revised manuscript has improved significantly, but there are still some areas that could benefit from further refinement. The main areas that need improvement include
Complex and Long Sentences - Some sentences are too long and difficult to follow. Simplifying them would improve readability and clarity.
Inconsistent verb tenses - Some sections shift inconsistently between present and past tense. Maintaining a consistent tense (usually past tense for describing completed experiments and present tense for general scientific facts) would improve flow.
Minor grammatical errors - While the major issues have been addressed, some minor grammatical errors remain, such as incorrect prepositions and word choices.
Overuse of jargon - Some sentences use too many complex technical terms in a row, making them difficult to understand. A slight simplification of the language would make the text more accessible to a broader audience.
Recommendation: A final round of professional editing by a native English speaker or language editing service would ensure that the manuscript meets high linguistic standards. Revisions should focus on sentence structure, verb consistency, and clarity without altering the scientific meaning.
Author Response
Authors Response for reviewers 1
Cover Letter and Author Response for Manuscript [Soil bulk density, aggregates, carbon stabilization, nutrients and vegetation traits as affected by manures gradients regimes under alpine meadows of Qinghai-Tibetan Plateau ecosystem].
March 22, 2025
The Revised areas are in blue font.
Dear respected editor,
We thank you and the reviewers for your encouraging comments on our manuscript entitled [Soil bulk density, aggregates, carbon stabilization, nutrients and vegetation traits as affected by manures gradients regimes under alpine meadows of Qinghai-Tibetan Plateau ecosystem]. We appreciate the opportunity to revise and improve this manuscript to a level suitable for publication. We carefully considered all reviewer comments and made point by point revisions to the manuscript to address these comments. Our responses to the reviewer comments are listed below in the blue font and are shown in blue font in the revised manuscript.
We hope our efforts have addressed all of the concerns to level required for publication. Thank you very much for your help. We look forward to hearing from you. Please contact us if there are questions or concerns.
Thank you very much.
Sincerely’
Authors:
Mahran Sadiq1,2,3,, Nasir Rahim2, Majid Mahmood Tahir2, Aqila Shaheen2, M Ajmal Ali3, Mohamed S. Elshikh3, Fu Ran1, Guoxiang Chen1 and Bai Xiaoming1,4*
1College of Grassland Science, Gansu Agricultural University, Lanzhou 730070, China; khanmahran420@gmail.com (M.S.)
2Department of Soil and Environmental Sciences, University of Poonch Rawalakot, AJK 12350, Pakistan; majidmahmood@upr.edu.pk (M.M.T.)
3Department of Botany and Microbiology, College of Science, King Saud University, Riyadh 11451, Saudi Arabia
4Key Laboratory of Grassland Ecosystem (Ministry of Education), Lanzhou 730070, China
*Correspondence: baixm@gsau.edu.cn
Responses to Reviewers Comments
After reviewing the revised manuscript "Soil bulk density, aggregates, carbon stabilization, nutrients and vegetation traits as affected by manure gradient regimes under alpine meadows of the Qinghai-Tibetan Plateau ecosystem", it is evident that the authors have made substantial improvements based on the reviewers' comments. The structure of the manuscript is now clearer, the introduction has been streamlined to focus on key research aspects, and the results section has been better organized into logical subsections, making it easier for the reader to follow the experimental results and their interpretation.
Response: Many thanks.
The scientific content is well presented and coherent, providing a comprehensive discussion of climate change impacts on meadow ecosystems, soil organic carbon (SOC) stabilization, vegetation structure, and nutrient dynamics under different organic manure gradients. The experimental section has been significantly improved, with clear justification for the selection of specific manure application rates (0.54, 1.08, 1.62, 2.16, and 2.70 kg/m²). In addition, the authors have provided a more detailed explanation of the randomized complete block design (RCBD), plot size, replication strategy, and analytical methods used to measure SOC fractions and nutrient levels.
Response: Bundle thanks for encouraging.
Data presentation has also been improved, with figures and tables more effectively referenced within the text. However, some figures still contain excessive information, making them somewhat complex to interpret at first glance. The simplification of figure annotations and captions would further improve clarity and accessibility.
Response: Now its modified according to respected reviewers suggestions. The captions and annotations are very very clear and simple. Its very easy for readers to understand in the future.
The manuscript has undergone significant grammatical and linguistic improvements, making it more readable. Ambiguous phrases have been corrected, and technical terminology is now more precise. However, some sections, particularly in the Results and Discussion, still contain long and complex sentences that could be further refined to improve readability.
Response: We respect the suggestion of respected reviewer, however, now Results and Discussion section has revised. All sentences are very clear with professional wordings. Each and every word that we have used in the manuscript is a professional and scientifically correct and belongs to specific soil and environmental sciences field, which increase the readability for coming scientists.
A key improvement is the expanded discussion of the negative response of the Carbon Management Index (CMI) in macroaggregates, with a clearer explanation of the underlying mechanisms and supporting references. The conclusion section has also been improved, with a stronger statement on the implications of the study for grassland management, environmental sustainability and climate change mitigation. However, highlighting specific recommendations for practical applications and future research directions would further strengthen this section.
Response: Now this section has revised and additions are in blue font in the text. Now, specific recommendations for practical applications and future research directions are mentioned in the manuscript.
The manuscript has been significantly improved and most of the reviewers' concerns have been adequately addressed. However, minor revisions are still needed, particularly to refine complex sentences, simplify data visualization, and further emphasize the practical implications of the study. I recommend publication after minor revisions focusing on language refinement, simplification of figures, and improved clarity of conclusions.
Comments on the Quality of English Language
The English in the revised manuscript has improved significantly, but there are still some areas that could benefit from further refinement. The main areas that need improvement include
Complex and Long Sentences - Some sentences are too long and difficult to follow. Simplifying them would improve readability and clarity.
Response: We respect the suggestion of respected reviewer. We have revised thoroughly. All sentences are clear and easy to understand for readability.
Inconsistent verb tenses - Some sections shift inconsistently between present and past tense. Maintaining a consistent tense (usually past tense for describing completed experiments and present tense for general scientific facts) would improve flow.
Response: Now its revised according to respected reviewers suggestions as the complete study was described in past tense.
Minor grammatical errors - While the major issues have been addressed, some minor grammatical errors remain, such as incorrect prepositions and word choices.
Response: Now its modified. All words are scientifically corrected and used in plenty of studies globally in the literature.
Overuse of jargon - Some sentences use too many complex technical terms in a row, making them difficult to understand. A slight simplification of the language would make the text more accessible to a broader audience.
Response: Now its revised. All words are very simple and easy to scientists to read.
Recommendation: A final round of professional editing by a native English speaker or language editing service would ensure that the manuscript meets high linguistic standards. Revisions should focus on sentence structure, verb consistency, and clarity without altering the scientific meaning.
Response: We carefully considered all reviewer comments and made point by point revisions to the manuscript to address these comments. We hope our efforts have addressed all of the concerns to level required for publication. Thank you very much for your help.
